# Direct conversion of various phosphate sources to a versatile P-X reagent [TBA][PO$_2$X$_2$] via redox-neutral halogenation

Yaling Tian [1,3], Dong-ping Chen[1,3], Yao Chai [1], Ming Li [1], Xi-Cun Wang [1], Zhengyin Du [1], Xiaofeng Wu [1,2] & Zheng-Jun Quan [1] ✉

Inorganic phosphates hold significant potential as ideal natural building blocks, forming a fundamental basis for organic and biochemical synthesis. However, their limited solubility, inherent chemical stability, and low reactivity pose substantial challenges to converting phosphates into organophosphates under mild conditions. This study introduces an efficient method for the direct conversion of phosphates into P(V)-X reagents, [TBA][PO$_2$X$_2$] (X = Cl, F), via a redox-neutral halogenation process. This method utilizes cyanuric chloride (or cyanuric fluoride) as the halogenation reagent, in combination with 1-formylpyrrolidine and tetrabutylammonium chloride (TBAC), under ambient conditions. The approach enables effective halogenation conversion for various P(V) sources, including orthophosphates, pyrophosphoric acid, Na$_3$P$_3$O$_9$ and P$_2$O$_5$. Furthermore, we demonstrate the synthetic utility of the P(V)-Cl reagent in the phosphorylation of diverse *O*-, *S*-, *N*- and *C*-nucleophiles. Key advantages of this conversion process include the use of inexpensive and readily available chemicals, the avoidance of high-energy redox reactions, and the generation of a reactive yet stable P(V)-X reagent.

Phosphorus is essential to the fundamental processes of life, playing a key role in nearly all biological activities[1-5]. For instance, organophosphorus compounds (OPCs) are essential components in natural systems[5], biosynthesis[6-8], and the formation of nucleic acid frameworks[8-11]. Furthermore, phosphorus-containing compounds have been widely utilized across various fields, including pharmaceuticals, food supplements, pesticides, flame retardants, electrolytes, and catalysts[12]. The synthesis of OPCs is primarily based on bulk chemicals such as PCl$_3$, PCl$_5$, POCl$_3$, P$_2$O$_5$, and PH$_3$, which are produced through the chlorination (or oxygenation) of white phosphorus (P$_4$) or via the hydrogenation of P$_4$ (thermal process) (Fig. 1a, b)[13-18]. These bulk chemicals pose significant environmental hazards due to their classification as dangerous volatile liquids and gases, as well as the cumbersome work-up processes they require. However, recent advancements have shown promise in developing safer, low-energy

methods for synthesizing P$_4$[19,20]. A more appealing approach involves directly functionalizing P$_4$ to convert it into phosphorus-containing chemicals, thereby avoiding the production of chlorinated bulk chemicals derived from P$_4$ (Fig. 1b)[21-30].

Phosphorus predominantly occurs in nature as phosphate rock minerals, existing in the most stable +5 oxidation state and widely recognized as pentavalent phosphorus (P(V)). Annually, ~90 million tons of these minerals are utilized for the industrial production of phosphoric acid through the wet process. Therefore, the development of efficient and cost-effective conversion techniques for affordable phosphates into stable phosphorus synthons under redox-neutral conditions is of utmost importance. However, the poor solubility and high stability of phosphate make it challenging for nucleophilic attacks on phosphorus atoms from both kinetic and thermodynamic standpoints, limiting the number of reagents that can effectively activate it.

[1]Gansu International Scientific and Technological Cooperation Base of Water-Retention Chemical Functional Materials, College of Chemistry and Chemical Engineering, Northwest Normal University, Lanzhou, Gansu, PR China. [2]Materials Innovation Factory, and Department of Chemistry, University of Liverpool, Liverpool L69 7ZD, UK. [3]These authors contributed equally: Yaling Tian, Dong-ping Chen. ✉e-mail: quanzhengjun@hotmail.com

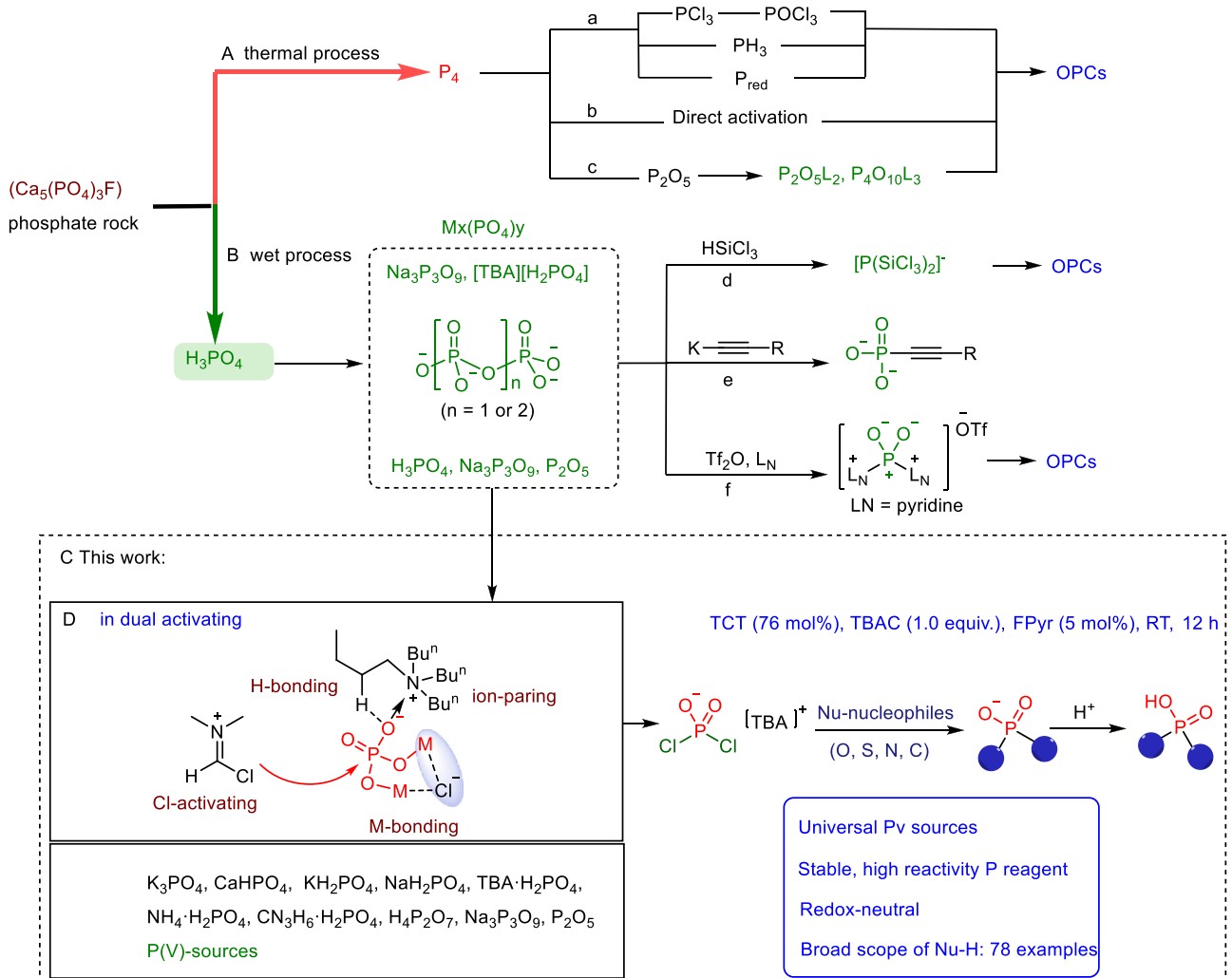

**Fig. 1 | Comparison of different pathways conversing phosphate salts to OPCs.**
A Synthesis of OPCs from $P_4$. **a** Prepared by bulk chemicals such as $PCl_3$, $PCl_5$, $POCl_3$, etc. **b** An alternative route to directly convert $P_4$ to OPCs. **c** Prepared by $P_2O_5L_2$ and $P_4O_{10}L_3$ resulted from $P_2O_5$. B Synthesis of OPCs from phosphate. **d** Prepared by valence-active intermediates from reducing phosphates. **e, f** Prepared by redox-neutral phosphate conversion strategy. C This work (gray box) presents an approach that converts phosphates into P–X reagents for synthesizing a variety of OPCs. A suggested mode of enhancing the solubility and reactivity of phosphate with TBAC is an example (D).

The specific P–O bond in $PO_4^{3-}$ necessitates harsh reaction conditions and is often associated with low reaction selectivity[31]. In earlier studies, orthophosphoric acid and its derivatives were employed as P(V) reagents in the synthesis of chemical oligonucleotides, enabling the formation of phosphate monoesters (including diesters) through esterification reactions[32–38]. Recently, methods for synthesizing OPCs through the reduction of phosphoric acid or condensed phosphates have been developed. In 2018, Cummins et al. achieved a significant milestone by preparing the bis(trichlorosilyl)phosphide anion via the reduction of trimetaphosphate or tetrabutylammonium phosphate ([TBA][$H_2PO_4$])[39,40]. This anion was then utilized as a phosphorus transfer reagent in the synthesis of various organophosphorus derivatives (Fig. 1d)[41–43]. In 2023, Cummins further investigated the reaction of $P_2O_5$ with specific N-donor bases to form adducts like $P_2O_5L_2$ and $P_4O_{10}L_3$ (L represents N-donor bases), which were effective conjugates in phosphorylation reactions[10,44]. Besides, limited approaches involving direct redox-neutral conversion of phosphates to OPCs are currently developing (Fig. 1c). A significant breakthrough occurred in 2023 when two independent groups, led by Cummins and Weigand, respectively, achieved notable progress in this area. Thus, Cummins et al.[45] developed the mechanochemical phosphorylation of acetylides with condensed phosphates, introducing a conversion strategy for

phosphates into OPCs (Fig. 1e). In parallel, Weigand et al.[46] reported an approach using trifluoromethanesulfonic anhydride (Tf$_2$O) and pyridine to directly convert P(V) sources into the versatile $PO_2^+$ phosphorylation agent (pyridine)$_2PO_2$[OTf]. Tf$_2$O is essential in breaking P–O bonds and stabilizing the resulting cationic P(V) center, with the aid of N-donor bases acting as effective electron donors. These methods provide redox-neutral access to a range of value-added P(V) chemicals downstream of low-cost phosphoric acid or other phosphate sources (Fig. 1f).

Inspired by the pioneering research of Cummins and Weigand, we propose that the solubility of phosphate salts can be significantly improved through the strategic application of an ion-exchange reagent[41]. Moreover, it is well established that phosphate can function as both a dual ion-pairing anion and a hydrogen-bonding acceptor in phosphoric acid-catalyzed reactions[47,48]. We hypothesized that enhancing the solubility and reactivity of phosphate salts could facilitate dual ion-pairing and hydrogen-bonding interactions between the cationic TBA and the $PO_4^{3-}$ anion (Fig. 1D). Furthermore, the P–O bond in the $PO_4^{3-}$ salt can be effectively activated by using reagents that selectively target hydroxyl groups, such as halogenating agents. Among these, cyanuric chloride (TCT, also known as trichlorotriazine) has been demonstrated to be one of the most

cost-effective reagents, particularly when combined with an amide, for activating the hydroxyl groups of alcohols and acids, offering a viable alternative to phosgene[49,50].

Here, we show a general chlorination reaction for phosphates using halogenation reagents such as TCT (or cyanuric fluoride), in combination with the catalyst 1-formylpyrrolidine (FPyr) and an activating agent like tetrabutylammonium chloride (TBAC). This strategy enhances the solubility and reactivity of phosphates, resulting in the formation of the P(V)−X reagent $[TBA][PO_2X_2]$ **1** (X = Cl or F), which maintains the oxidation state of the phosphorus atom. The chlorination reaction shows excellent scope and compatibility with various P(V)-sources. We further demonstrate that the P(V)-Cl reagent serves as a highly effective phosphorylation reagent, enabling the synthesis of a variety of OPCs (Fig. 1C).

## Results and discussion

### Synthesis of P(V)−X reagents $[TBA][PO_2X_2]$ (X = Cl, F)

Phosphoryl dichloride $(KPO_2Cl_2)$ is typically synthesized by reacting $POCl_3$ with $KHCO_3$ in an aprotic solvent. The corresponding fluoride $KPO_2F_2$ is then obtained through a fluoride-chloride exchange reaction of $KPO_2Cl_2$[51]. Our studies commenced with the conversion of the known $[TBA][H_2PO_4]$ as a model P(V) source into the phosphorodichloridate product $[TBA][PO_2Cl_2]$ (**1a**) (see Supplementary Information S.2.1, Table S1). By optimizing the reaction conditions $[TBA][H_2PO_4]$ reacted with TCT (76 mol%) in the presence of FPyr (5 mol%) at room temperature for 12 h, resulting in an isolated yield of 96% for product **1a**. Furthermore, substituting FPyr with 5 mol% DMF resulted in a yield reduction of 67%. The absence of an amide catalyst led to a significant drop in yield, highlighting the crucial role of the catalyst in activating the C-Cl bonds in TCT and improving overall reaction efficiency. When investigating alternative chlorination reagents such as benzoyl chloride, trichloroisocyanuric acid, phosphorus pentachloride $(PCl_5)$, N-chlorosuccinimide and triphosgene (BTC), it was found that these reagents were less efficient than TCT (Fig. 2a). The formation of $[PO_2Cl_2]^-$ (signal, δ = −6.77 ppm), as evidenced by $^{31}P$ NMR spectroscopy of the reaction mixture. An intermediate was also observed (δ = −28 ppm) and this intermediate subsequently underwent further transformations to yield product **1a** (Fig. 2d). Unfortunately, isolating the intermediate at −28 ppm is not feasible (refer to Supplementary Information S.4.1, Tables S19 and S21). This finding is consistent with previously reported results[51,52]. The structure of **1a** was further verified by X-ray crystallography (See Supplementary Information S.7, Fig. S15). To evaluate the scalability of $[TBA][PO_2Cl_2]$, a 100 mmol scale experiment was carried out, which yielded **1a** in 90% yield (33.7 g). Additionally, a solid precipitate formed during the reaction, which was isolated by filtration and dried, yielding 94% of the theoretical amount of cyanuric acid (CA) (9.28 g). CA can be further converted to TCT using the chlorination reagents $POCl_3$[53] and $SOCl_2$ (refer to Supplementary Information S.4.3, Table S22).

With the optimized reaction conditions, we proceeded to explore the scope of the chlorination reaction using commercially available phosphate substrates. However, when orthophosphates such as $CaHPO_4$ and $K_3PO_4$ were employed, no reaction occurred under the optimal conditions. This lack of reactivity can be attributed to the low solubility of these phosphates in organic solvents. To our delight, the addition of a stoichiometric amount of TBAC resulted in a clear solution and significantly enhanced the reaction activity, leading to the full formation of **1a**. In contrast, substituting TBAC with tetramethylammonium (TMA) salts, such as TMAC and TMAB, halted the reaction entirely. These results underscore the crucial role of the TBA salt in facilitating the chlorination process. As shown in Fig. 2b, commercially available phosphates, such as orthophosphates $(K_3PO_4,$ $KH_2PO_4,$ $NaH_2PO_4,$ $[TBA]H_2PO_4,$ $[NH_4]H_2PO_4,$ $[CN_3H_6]H_2PO_4),$ pyrophosphoric acid $(H_4P_2O_7),$ polyphosphates $(Na_3P_3O_9)$ and phosphorus pentoxide $(P_2O_5),$ were successfully transformed into phosphorodichloridate **1a** by TCT and FPyr in the presence of TBAC

(Fig. 2b). It is noteworthy that the dichlorophosphoryl product $[CN_3H_6][PO_2Cl_2]$ **1b** (Fig. 2b), derived from guanidine phosphate $([CN_3H_6]H_2PO_4),$ can be obtained with an impressive yield of 86% even in the absence of TBAC. In this reaction, guanidinium functions similarly to the TBA cation, effectively promoting the reaction.

Encouraged by the promising results of the chlorination conversion, we extended our investigation to explore the fluorination of phosphates (refer to Supplementary Information S.2.2). By using cyanuric fluoride as the fluorine source for the direct fluorination of $[TBA][H_2PO_4],$ we achieved the phosphorodifluoridate **1c** with an 88% isolated yield under the standard conditions (Fig. 2d). Among the fluorinating reagents examined, cyanuric fluoride demonstrated superior performance compared to other agents, such as diethylaminosulfur trifluoride (DAST), (diethylamino)difluorosulfonium tetrafluoroborate (XtalFluor-E), and various "F-Reagents," which produced lower yields or only trace amounts of **1c**. The fluorination of various phosphates, including $K_3PO_4,$ $NH_4(H_2PO_4),$ pyrophosphoric acid, $KH_2PO_4,$ $NaH_2PO_4,$ $CaHPO_4$ and $P_2O_5,$ was successfully achieved with good yields (Fig. 2c, f). Additionally, the reaction of $Na_3P_3O_9$ with cyanuric fluoride led to the formation of monofluorinated compound $[TBA][HPO_3F]$ **1d** was obtained separately (Fig. 2c). An alternative one-pot F−Cl exchange reaction was also explored, yielding compound **1c** when $[TBA][H_2PO_4]$ was reacted with TCT and TBAF·$3H_2O$ in a single-step process (refer to Supplementary Information S.2.2, Table S8).

The halogenation of phosphates generated solid reagents, $[TBA][PO_2Cl_2]$ and $[TBA][PO_2F_2],$ which are stable under ambient conditions. This contrasts sharply with conventional toxic gases and moisture-sensitive liquid phosphate halides (such as $PCl_3,$ $POCl_3,$ $POF_3,$ $PSF_3,$ and $PF_5),$ highlighting the practical advantages of these solid-phase reagents. Stability experiments reveal that $[TBA][PO_2Cl_2]$ rapidly hydrolyzes in water to form phosphoric acid, however, it exhibits enhanced stability when exposed to air or dissolved in organic solvents, where it dimerizes very slowly. In a sealed environment, it remains stable for up to 30 days without any signs of decomposition. In comparison, $[TBA][PO_2F_2]$ exhibits significantly enhanced stability compared to $[TBA][PO_2Cl_2]$ (refer to Supplementary Information, Table S11). Notably, it remains stable not only in organic solvents but also exhibits exceptional stability in water. Analysis of the P−Cl and P−F bond lengths reveals that the P−Cl bonds are weaker (~2.0 Å), which may facilitate simultaneous reactions, in contrast to the thermodynamically stable P−F bonds (around 1.5 Å). These findings suggest that P−X (where X = F, Cl) bonded compounds are reactive yet stable reagents suitable for the synthesis of OPCs (refer to Supplementary Information S.2.4).

### Synthetic application of P(V)−X reagent

Building on the successful large-scale preparation of the P(V)−X reagent $[TBA][PO_2X_2],$ we further explored the utility of $[TBA][PO_2Cl_2]$ in synthesizing complex OPCs. Our investigation commenced with the phosphorylation of O-nucleophiles (**2**) using **1a**, as organophosphoric acids are among the most widely utilized ligands[54]. Initially, the phosphorylation reaction between **1a** and binaphthol was conducted in the presence of NaOH and $Na_2S_2O_3$·$5H_2O,$ using THF as the solvent at 40 °C for 1 h. This reaction yielded the organophosphate ester **I-1** with a 76% isolated yield. The structure of the product was confirmed through single-crystal X-ray diffraction analysis (Fig. 3). The experiments demonstrated that the addition of $Na_2S_2O_3$·$5H_2O$ significantly enhances the reactivity and improves the conversion rate (refer to Supplementary Information S5). We propose that the acceleration of the alcoholysis reaction by $Na_2S_2O_3$·$5H_2O$ is due to a unique interaction between the S(VI) center and the Cl⁻ anion, which weakens the P−Cl bond and makes it more susceptible to nucleophilic attack.

The derivatization experiments confirmed the successful reaction of **1a** with chiral binaphthol, leading to the formation of chiral binaphthol phosphates (Fig. 3, **I-2, I-3**). Further investigations

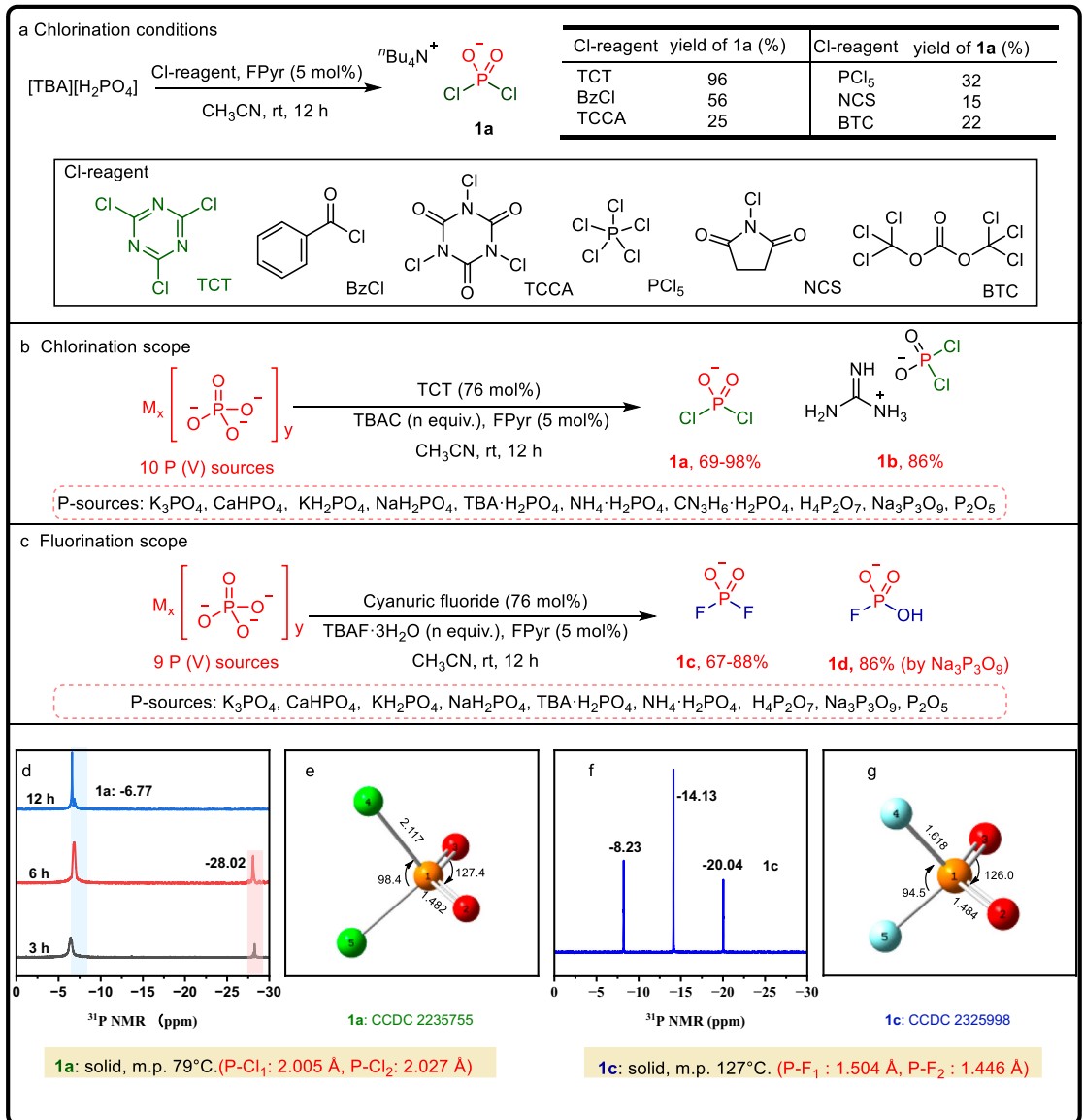

**Fig. 2 | Preparation of P(V)−X reagents. a** Comparison of synthetic **1a** by using different chlorination reagents. **b** The scope of different P(V)-sources in chlorination reaction. **c** The scope of different P(V)-sources in fluorination reaction. **d** ³¹P NMR monitored the reaction process of **1a**. **e** Crystal structural model and physical property of **1a**. **f** ³¹P NMR of **1c** [TBA][PO₂F₂]. **g** Crystal structural model and physical property of **1c**. Reaction conditions for **1a**: [TBA][H₂PO₄] (3 mmol), TCT (76 mol%), FPyr (5 mol%), CH₃CN (2 mL), rt, 12 h. (The amount of TBAC is equal to the amount of phosphorus atom in the corresponding phosphate, refer to Supplementary Information S.2.1, Table S2). Reaction conditions for **1b**: Guanidine phosphate (3 mmol), TCT (76 mol%), FPyr (5 mol%), CH₃CN (2 mL), rt, 12 h. Reaction conditions for **1c**: [a] Direct fluorination. [TBA][H₂PO₄] (3 mmol), cyanuric fluoride (76 mol%), FPyr (5 mol%), CH₃CN (2 mL), rt, 12 h. [b] F−Cl exchange. [TBA][H₂PO₄] (3 mmol), TCT (76 mol%), TBAF 3H₂O (2.0 eq), CH₃CN (2 mL), rt, 12 h. (The amount of TBAF 3H₂O is equal to the amount of phosphorus atom in the corresponding phosphate, refer to Supplementary Information S.2.2, Table S6). [c] Reaction conditions for **1d**: Na₃P₃O₉ (3 mmol), cyanuric fluoride (2 eq), TBAF 3H₂O (3.0 eq), FPyr (5 mol%), CH₃CN (2 mL), rt, 12 h.

broadened the scope by synthesizing seven-membered phosphorus heterocyclic compounds (**I-4**−**I-6**) from bisphenol substrates. These compounds exhibit structural similarities to known organophosphorus ligands, indicating their potential applications in catalytic organic reactions. Additionally, a six-membered cyclic phosphorus compound, 4H-benzo[d][1,3,2]dioxaphosphinin-2-olate 2-oxide (**I-7**), was successfully synthesized with a yield of 76% using o-hydroxybenzyl alcohol as the substrate. Further studies were conducted to assess the scalability of phenolic substrates. The introduction of either electron-donating groups (such as Me and MeO) or electron-withdrawing groups (including Cl, Br, CN, NO₂, and Ph) on the phenol resulted in the formation of corresponding phosphodiester compounds (**I-8**−**I-14**), achieving medium to high yields. The synthesis of compounds (**I-15**, **I-16**) was successfully accomplished using F- and CF₃-substituted phenols. The application of meta-substituted phenols, such as Br or NO₂, facilitated the production of the desired phosphodiester (**I-17**, **I-18**). Substrates containing multiple electron-withdrawing groups proved to be compatible with this reaction. For instance, bis(2,4-dichlorophenyl) phosphate (**I-19**) and bis(3,4,5-trifluorophenyl) phosphate (**I-20**) were synthesized successfully. Additionally, the use of naphthol, as well as dibromo- and aldehyde-substituted naphthols, led to the formation of di(naphthalene-2-yl) phosphates (**I-21**−**I-23**). It is noteworthy that the synthesis of pyridine phosphates has been challenging, with few reports on these compounds to date[55,56]. We successfully synthesized unique trifluoromethyl- and nitro-substituted bis(pyridin-2-yl) phosphates (**I-24**, **I-25**). The versatility of the reaction

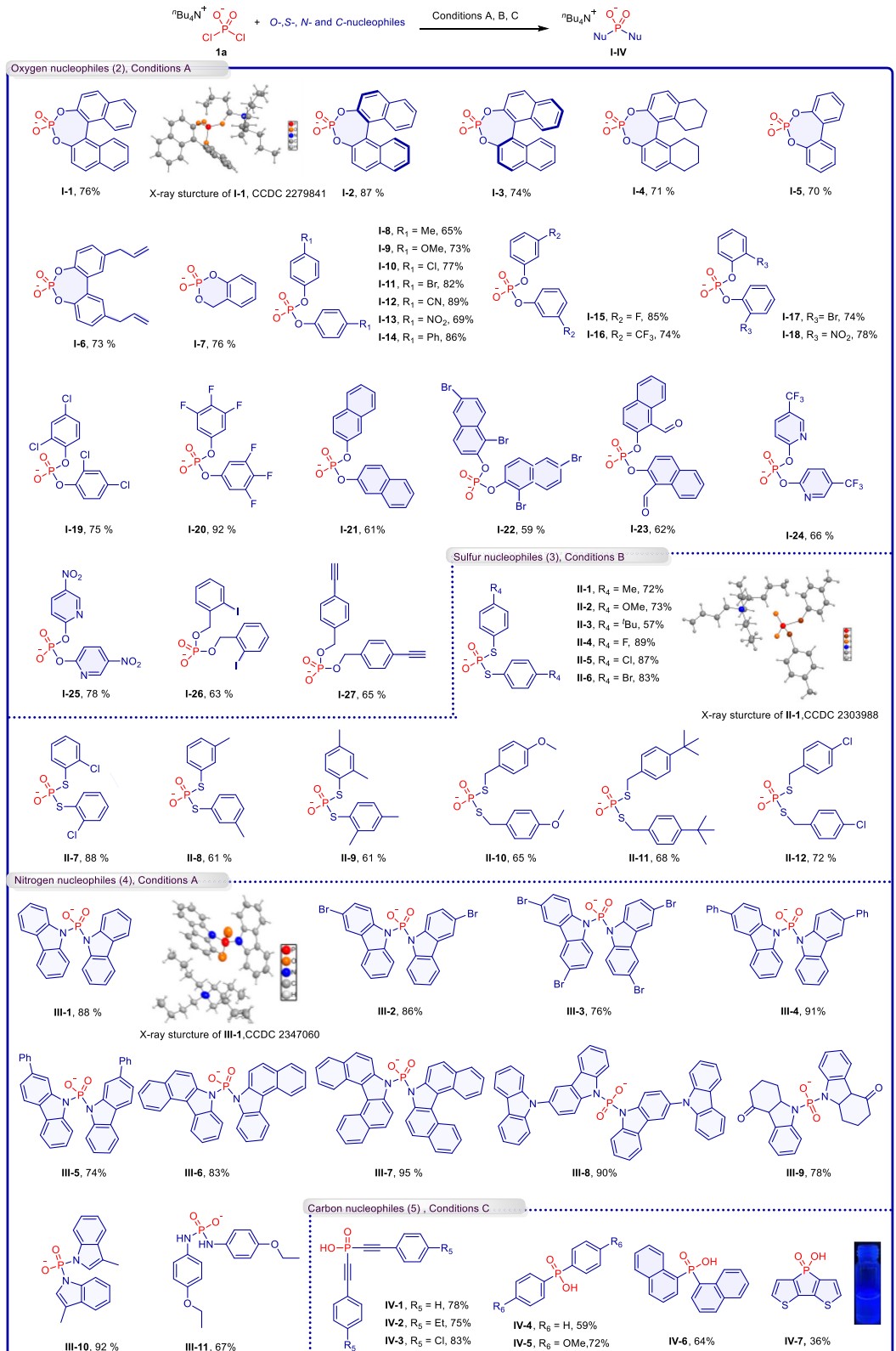

**Fig. 3 | Substrate scope of phosphorylation of *O*-, *S*-, *N*- and *C*- nucleophiles with [TBA][PO₂Cl₂].** For simplicity, the *ⁿ*Bu₄N⁺ group in the structural formula of the products has been omitted. **Conditions A**: **1a** (0.2 mmol), *O*- and *N*-nucleophiles (2.5 eq., 0.5 mmol), NaOH (2.5 eq., 0.5 mmol), Na₂S₂O₃.5H₂O, (2.5 eq., 0.5 mmol), THF (1 mL), 40 °C, 1–3 h. **Conditions B**: **1a** (0.2 mmol), *S*-nucleophiles (2.5 eq., 0.5 mmol), NaOH (2.5 eq., 0.5 mmol), THF (1 mL), rt, 5 min. **Conditions C**: **1a** (0.2 mmol), *C*-nucleophiles (2.5 eq., 0.5 mmol), THF (1 mL), −78 °C - rt, 3 h.

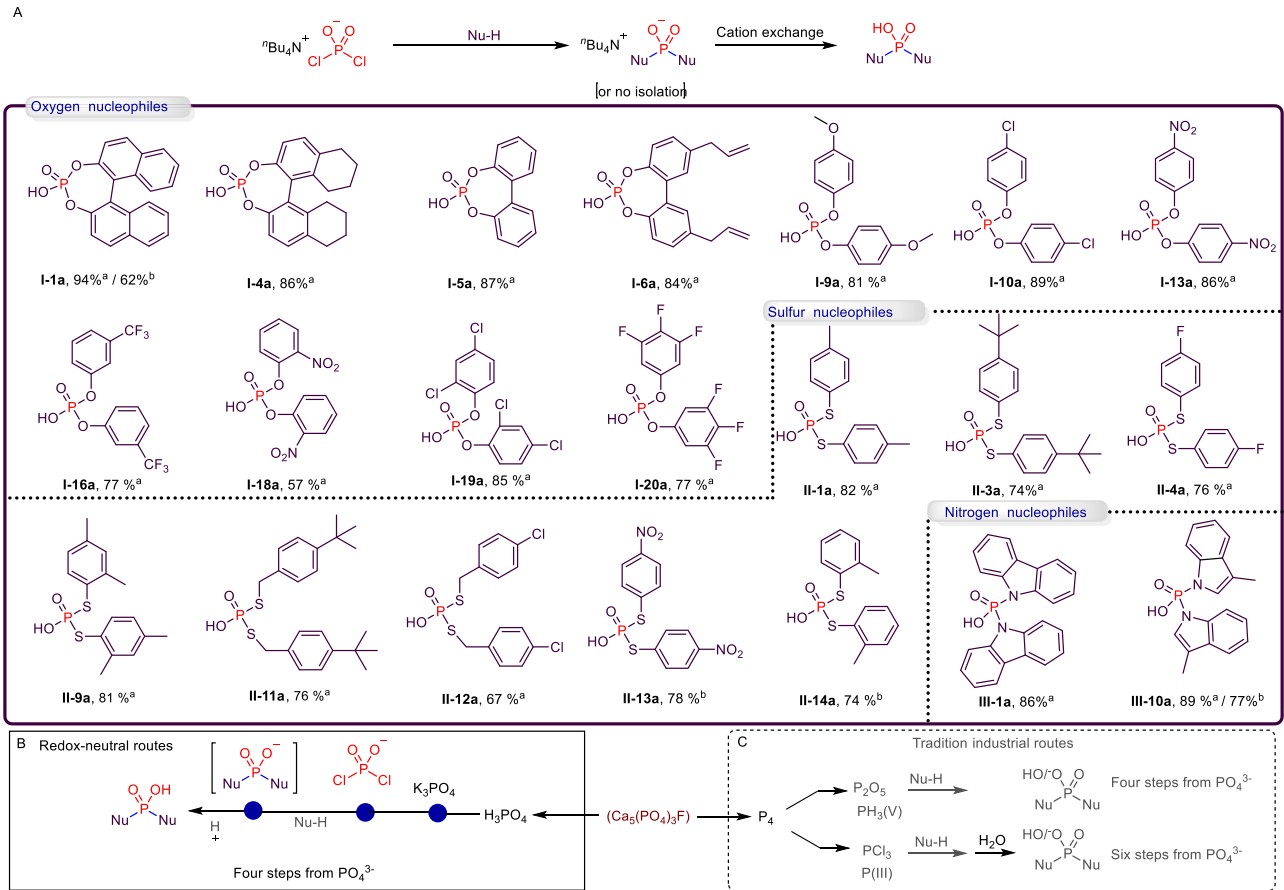

**Fig. 4 | Preparation of phosphoric acids. A** Cation exchange of organophosphate salts. [a]The yields are those of stepwise reactions. [b]The yields are the reactions in one pot. Comparison of this method (**B**) to traditional industrial routes (**C**).

with various functional groups was further demonstrated by using benzyl alcohol substrates, which led to the formation of products (**I-26**, **I-27**). These products serve as excellent building blocks containing alkynyl and iodine motifs, enabling further structural diversification through metal-catalyzed coupling reactions.

Sulfur-containing OPCs have significant applications in the fields of biochemistry and agrochemistry[57,58]. By adjusting the optimal conditions to the absence of $Na_2S_2O_3 \cdot 5H_2O$ (conditions B), we successfully synthesized a diverse array of dithiol-phosphonate salts (**II**) through the reaction of **1a** with various thiol substrates (**3**) (Fig. 3). During the substrate expansion, we observed that the reaction proceeded successfully regardless of the electron-withdrawing or electron-donating groups on the substrate, resulting in the formation of dithiophosphate salts (**II-1**–**II-6** and **II-7**–**II-9**). When benzyl mercaptan was employed, the reaction still yielded the corresponding products (**II-10**–**II-12**), although with slightly lower yields compared to other thiol substrates.

Carbazolyl phosphorus compounds are widely utilized in the field of optoelectronic materials. However, the existing synthesis methods for these substrates often involve the use of *n*-butyllithium and are frequently associated with considerable complexity, thereby hinders further exploration of analogous structures[59,60]. we conducted further research on the phosphorylation of carbazole and indole (**4**) using **1a** (Fig. 3). Under condition A, the reaction of **1a** with carbazole over a period of 3 h successfully produced carbazolyl phosphorus (**III-1**). When carbazoles were substituted with (bis)bromine or a phenyl group, phosphoryl carbazole compounds (**III-2**–**III-5**) were generated. Fused cyclic carbazole substrates also demonstrated reactivity, resulting in products (**III-6**, **III-7**) with good to excellent yields. Notably, the carbazole-substituted carbazole also produced the

corresponding product (**III-8**). Of particular significance, we successfully employed the drug intermediate 2,3-dihydrocarbazol-4(1H)-one[61] to yield the corresponding phosphoramide product (**III-9**). Furthermore, the reaction of **1a** with indole was successfully achieved, resulting in the formation of phosphorylated indole (**III-10**) in 92% yield. Similarly, the *p*-ethoxy-substituted aniline underwent a reaction to produce the desired phosphoramidite product (**III-11**).

To further investigate the potential of *C*-nucleophiles (**5**) in this phosphorylation, the range of nucleophiles was expanded to alkyne and aryl lithium reagents. This revealed that **1a** successfully reacts with organolithium reagents forming P–C bonds for the synthesis of dialkynyl hypophosphorous acid (**IV-1**–**IV-3**) and diaryl hypophosphorous acid (**IV-4**–**IV-7**). It is noteworthy that compound **IV-7** exhibits fluorescence at 365 nm (Fig. 3).

Following the successful synthesis of phosphorylated salts, an investigation was conducted into the acidification reaction of the obtained phosphate salts (**I-III**). Consequently, a series of acidified phosphorylated products were synthesized. For example, phosphoric acids (**I-1a**, **I-4a-6a**, **I-9a**, **I-10a**, **I-13**, **I-16a**, **I-18a-I-20a**), (**II-1a**, **II-3a**, **II-4a**, **II-9a**, **II-11a-14a**), indolyl and carbazolyl phosphoric acids (**III-1a**, **III-10a**) were obtained (Fig. 4). It is noteworthy that this method allows for the synthesis of a diverse range of products from phosphates in a mere two steps. This streamlined process reduces the number of reaction steps and offers cost savings in comparison to traditional industrial synthesis methods, which usually involve at least 4 or 6 steps. The utility of this reactivity is further enhanced when the phosphorylated products obtained do not require isolation. To investigate this possibility, we selected phosphorylated salts **I-1**, **II-13**, **II-14**, and **III-1**, which had not been isolated and treated them in situ

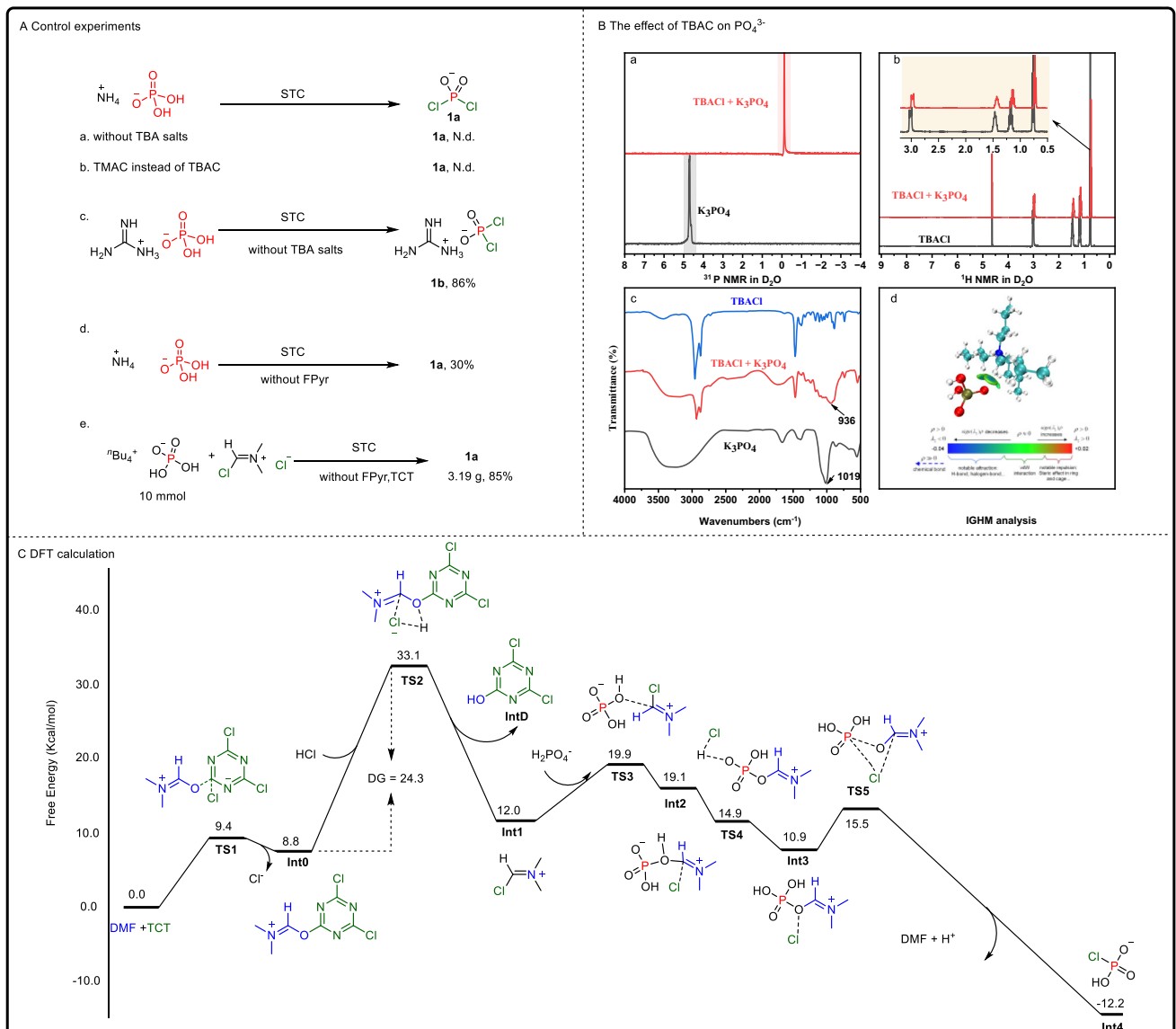

**Fig. 5 | Mechanism verification. A** Control experiments. Standard conditions (STC): [NH$_4$][H$_2$PO$_4$] (3 mmol), TCT (76 mol%), FPyr (5 mol%), TBA salts (1.0 equiv.) CH$_3$CN (2 mL), rt, 12 h. a Without TBA salts. b TMAC instead of TBAC. c Guanidine phosphate instead of [NH$_4$][H$_2$PO$_4$] in the absence of TBAC. d Without FPyr. e Synthesis **1a** using Vilsmeier reagents. **B** The effect of additive TABC on K$_3$PO$_4$. a The effect of TBACl on the $^{31}$P NMR of K$_3$PO$_4$ was studied. b The comparison of $^1$H NMR of TBAC and TBAC + K$_3$PO$_4$ mixed systems. c IR spectra of TBACl and TBAC + K$_3$PO$_4$ mixed system. (Display the peak of the phosphorus-oxygen bond). d IGHM analysis of [TBA][H$_2$PO$_4$] (red is "O" atom, brown is "P" atom, green is "C" atom, blue is "N" atom, gray is "H" atom). **C** Computed the favorable free energy profiles for the tentative reaction pathways.

with a cation exchange resin, resulting in the corresponding phosphoric acids **I-1a**, **II-13a**, **II-14a**, and **III-1a**. These findings reinforce the feasibility of synthesizing various OPCs using [TBA][PO$_2$Cl$_2$], thereby demonstrating the versatility and potential of this approach.

## Mechanism verification

To gain deeper insights into this redox-neutral chlorination process, a series of mechanistic experiments were conducted. We first evaluated the role of tetrabutylammonium (TBA) cations on phosphate solubility, using NH$_4$H$_2$PO$_4$ as a control. The results showed that NH$_4$H$_2$PO$_4$ did not produce the expected chlorination products under standard reaction conditions (Fig. 5A). Further investigations compared the reactivity of TBA versus TMA salts. However, no products were observed when TMA salts were used instead of TBA (controls a, b). Notably, a soluble guanidine phosphate salt produced the dichlorophosphoryl product **1b** in 86% yield, even in the absence of TBAC (control c).

To further probe the effect of TBAC in the reaction system, K$_3$PO$_4$ was used as a model compound, and the reaction system was monitored *via* $^{31}$P NMR and infrared spectroscopy (IR). The NMR results revealed that the addition of TBAC induced a high-field shift in both phosphorus and hydrogen NMR spectra. Similarly, IR spectroscopy showed a corresponding redshift (Fig. 5Ba–c). Independent gradient model (IGHM)[62] analysis of [TBA][H$_2$PO$_4$] using Multiwfn[63] and VMD[64] suggested a weak interaction between the CH hydrogens of TBA and H$_2$PO$_4^-$ anions (Fig. 5Bd). These findings indicate that TBAC may facilitate ion-exchange, ion-pairing, hydrogen bonding, and van der Waals interactions between [TBA]$^+$ and PO$_4^{3-}$, collectively activating the phosphate and driving the formation of product **1a**[65–67].

A crucial step in the chlorination process is hypothesized to involve the activation of tricyanuric chloride (TCT) by 1-formylpyrrolidine (FPyr), generating highly reactive Vilsmeier intermediates, which are essential for the successful progression of the reaction. In the absence

**Fig. 6 | Possible mechanism.** A plausible route for the conversion of [TBA][H₂PO₄] to **1a**. **A**: Vilsmeier reagent, **B** zwitterionic phosphate, **C** monochlorophosphate, **D** 4,6-dichloro-1,3,5-triazin-2-ol or 6-chloro-1,3,5-triazine-2,4-diol.

of FPyr, the chlorination reaction yielded only 30% of **1a** (control d). A scaled-up reaction using the preformed Vilsmeier reagent and [TBA][H₂PO₄] resulted in an 85% yield of **1a** (control e). Additionally, high-resolution mass spectrometry (HRMS) detected a zwitterionic intermediate **B** in the chlorination reaction of [TBA][H₂PO₄], with a [M + K]⁺ peak at *m/z* 217.9979 (Fig. 6). These results underscore the catalytic role of FPyr in generating the reactive Vilsmeier intermediates that are pivotal for the chlorination process. (control c).

Density functional theory (DFT) calculations were utilized to investigate the reaction mechanisms, with the detailed information provided in Fig. 5C. Since Vilsmeier aluminum is the most critical intermediate in the chlorination process, the DFT calculations primarily focus on the formation of Vilsmeier reagent (**Int1**) and the generation of the key monochloride-substituted intermediate (**Int4**). We also considered the important intermediates **Int2** and **Int3**. The formation of **Int1** occurs with an energy barrier of 24.3 kcal/mol (**TS(1,2)**), while the formation of **Int4** requires only a minimal energy input of 4.6 kcal/mol. Subsequently, **Int4** undergoes further chlorination to yield the target product **1a**.

Referring to the chlorination reaction reported for TCT[50] and based on the DFT results, we propose a possible mechanism for the chlorination of phosphates (Fig. 6). Initially, the solubility and reactivity barriers of phosphate salts in the transformation from P–OH to P–Cl can be overcome through a TBA-activated process. A critical subsequent step involves an amide (either DMF or FPyr) attacking the TCT, leading to the formation of Vilsmeier reagent **A**, which then rapidly reacts with phosphate to generate zwitterionic intermediate **B**, detected by HRMS (*m/z* 217.9979). Intermediate **B** is chlorinated with Cl⁻ ions to produce monochlorophosphate **C**. **C** is further chlorinated to obtain the product **1a**. Although **C** was not isolated likely due to its high reactivity, it was detected by HRMS ([M + H]⁺, *m/z* 114.9353), and its analogs monofluorophosphate **1c** was successfully isolated during the fluorination process.

This study addresses the limited methods available for the direct conversion of phosphates into P(V)–X reagents, such as [PO₂X₂]⁻, by employing cost-efficient commodity chemicals under ambient conditions. By focusing on the conversion of readily available phosphate salts into P(V)–X reagents, we establish these compounds as stable and

versatile precursors for synthesizing P(V)-containing species. The findings demonstrate that multiple universal phosphorus sources can be efficiently converted into the P(V)–X reagents ([TBA][PO₂X₂]). These reagents exhibit high reactivity and stability, making them valuable and advantage precursors in phosphorylation reactions, especially in contrast to many currently known phosphorylation reagents that are air- and/or moisture-sensitive or toxic. Overall, the current work introduces a practical strategic approach for converting simple phosphates into phosphorylation reagents, opening exciting prospects for the broad application of P(V) sources in the chemical synthesis of OPCs.

## Methods

### General procedure for the chlorination and fluoridation of phosphate

**General procedure for the preparation of [TBA][PO₂Cl₂].** A solution of [TBA][H₂PO₄] (3 mmol) and TCT (76 mol%) in CH₃CN, FPyr (5 mol%) was added into a round-bottom flask equipped with a magnetic stirring bar, and the reaction mixture was stirred at room temperature for 12 h. The insoluble material was directly filtered off, and the crude product was recrystallized from DCM and petroleum ether to yield the desired chlorinated product, [TBA][PO₂Cl₂].

### General procedure for the preparation of [TBA][PO₂F₂]

**Preparation of [TBA][PO₂F₂] using CaHPO₄ as a phosphine source.** A solution of the CaHPO₄ (3 mmol) and TBAF 3H₂O (3 mmol, 1.0 eq.) was added into a round-bottom flask equipped with a magnetic stirring bar at 0 °C in CH₃CN (2 mL), followed by the slow addition of DAST (3 mmol, 1.0 eq.) into it and stirred vigorously. After the addition, the temperature was raised to room temperature and the reaction was continued for 12 h. The precipitate was removed by filtering. The obtained filtrate was concentrated in vacuo and the resulting crude product was purified by recrystallization with DCM and petroleum ether to provide product [TBA][PO₂F₂] **1c** as a white solid.

**Preparation of [TBA][PO₂F₂] using [TBA][H₂PO₄] as a phosphine source.** A solution of [TBA][H₂PO₄] (3 mmol), cyanuric fluoride (2.28 mmol, 76 mol%), and FPyr (0.15 mmol, 5 mol%) was

added into a round-bottom flask equipped with a magnetic stirring bar in $CH_3CN$ (2 mL) and then stirred at room temperature for 12 h. The precipitate was removed by filtration, and the filtrate was concentrated under reduced pressure. The crude product was purified by recrystallization from DCM and petroleum ether to yield [TBA][$PO_2F_2$] **1c** as a white solid.

**General procedure for phosphorylation of O-nucleophile.** A solution of [TBA][$PO_2Cl_2$] (0.2 mmol), bisnaphthol (0.5 mmol, 2.5 eq.), NaOH (0.5 mmol, 2.5 eq.), and $Na_2S_2O_3 \cdot 5H_2O$ (0.5 mmol, 2.5 eq.) was added into a round-bottom flask equipped with a magnetic stirring bar in THF (1 mL) and stirred at 40 °C for 1 h. The aqueous layer was extracted with DCM (3 × 10.0 mL). The combined organic extracts were dried over anhydrous sodium sulfate and concentrated under reduced pressure. The residue was purified by column chromatography (silica gel, eluent DCM: MeOH = 100:1–20:1) to yield the desired product **I-1**.

**General procedure for cation exchange reactions.** A solution of product **I**, **II**, or **III** (0.2 mmol) and cation exchange resin (DOWEX 50 WX8) (15 mg) in acetone (2 mL) was stirred at room temperature for 30 min. The resin was removed by filtration, and the filtrate was concentrated under reduced pressure. The crude product was purified by recrystallization from DCM and petroleum ether to yield products **I-a**, **II-a**, or **III-a** as white solids.

**General procedure of [TBA][$PO_2Cl_2$] reaction with carbon nucleophiles**
**For example, the synthesis of IV-1.** Under an inert atmosphere, a solution of phenylacetylene (0.5 mmol) in THF (2 mL) was added into a round-bottom flask equipped with a magnetic stirring bar and then an equivalent amount of n-butyllithium (nBuLi) (1 M in diethyl ether) was added dropwise at 0 °C, and the mixture was vigorously stirred for 2 h. Subsequently, [TBA][$PO_2Cl_2$] (0.2 mmol) was introduced into the reaction mixture, and the reaction system was allowed to warm to room temperature, where it was further stirred for an additional hour. A saturated aqueous solution of ammonium chloride (10.0 mL) was then added. The aqueous layer was extracted with DCM (3 × 10.0 mL). The combined organic extracts were dried over anhydrous sodium sulfate and concentrated in vacuo. The residue was purified by column chromatography on silica gel, using a DCM:MeOH (10:1) eluent, to yield the desired product **IV-1**.

## Data availability
All data supporting the findings of this study are available within the article and its Supplementary Information files. All data are available from the corresponding author upon request. Crystallographic data for the structures reported in this Article have been deposited at the Cambridge Crystallographic Data Center, under deposition numbers CCDC 2235755 (**1a**), CCDC 2325998 (**1c**), CCDC 2279841 (**I-1**), CCDC 2303988 (**II-1**), CCDC 2347060 (**III-1**). Copies of the data can be obtained free of charge via https://www.ccdc.cam.ac.uk/structures/. Source data is provided with this article. Source data are provided with this paper.

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

## Acknowledgements

The authors acknowledge funding from the National Nature Science Foundation of China (No. 22061038, 22067018, 21562036), Foundation

of Northwest Normal University (No. NWNU-LKZD2023-04), Foundation of Gansu Provincial Department of Education (No. 2024CYZC-08), and Gansu Major Science and Technology Foundation (No. 25ZDWA008).

## Author contributions

Z.-J.Q. conceived the project. Y.L.T. carried out the experimental works and characterized the isolated products. Y.L.T. and Y.C. performed the control experiments and characterizations and collected the data. D.-P.C. performed the DFT calculations. M.L., Y.L.T., and X.-C.W. drafted the initial manuscript. Z.-J.Q. and X.F.W. revised the manuscript thoroughly. Z.-Y.D. reviewed all the content of the manuscript and supporting materials. All authors interpreted the data and contributed to the preparation of the manuscript.

## Competing interests

The authors declare no competing interests.
