## [Transparent Peer Review file · Nature Communications]

Direct conversion of various phosphate sources to a versatile P-X reagent [TBA][PO₂X₂] via redox-neutral halogenation

Corresponding Author: Professor Zheng-Jun Quan

Version 0:

Reviewer comments:

Reviewer #1

(Remarks to the Author)

This manuscript draft by Quan and colleagues describes the activation of various phosphate sources to dichlorophosphate anions, enabling subsequent functionalization with a range of O-, S-, N-, and C-centered nucleophiles under cleavage of the P-Cl bonds. This is achieved with cyanuric chloride as a chlorinating agent, 1-formylpyrrolidine as a catalyst, and tetrabutylammonium cations for increased solubility of the phosphates, identified as the preferred set of reagents from an extensive reaction screening. The authors further expand this methodology to include fluoride as an alternative leaving group. Additionally, mechanistic studies on the chlorinating activation step are presented.

Overall, I think that this work is of great interest to the scientific community in principle. The synthetic methodology is very simple and opens up an attractive alternative route to many organophosphorus compounds, which is underpinned by a large scope of exemplary target compounds. However, the scholarly representation of these results in the manuscript in its current form is insufficient and not suitable for publication. Because of this, I cannot recommend publication of this manuscript in this journal (or any other journal) in its current form and would advise the authors to significantly revise both their manuscript and supporting information, as many of these issues are quite obvious and could have been resolved long before submission. I have listed some exemplary concerns below.

In addition to this, I would like to leave one critical comment on pre-print publications because this work is already online (<https://www.researchsquare.com/article/rs-4564335/v1>) with a DOI number (<https://doi.org/10.21203/rs.3.rs-4564335/v1>). Publishing in pre-print journals without any peer review is dangerous, and this trend is, in my opinion, just the wrong direction but of course my personal view! This current work is unfortunately a showcase for it, as the scholarly representation and the conclusions derived from the results are not adequate and not by any chance suitable for publication.

1. The references given in the manuscript, especially in the introduction, are in many cases not appropriate for the statements they are meant to support. To give an example, the very first reference to support the importance of phosphorus in biological systems is a medicinal review on the role of phosphate in kidney diseases. Using such a specific reference for a very general point seems completely arbitrary to me, and the authors should choose their references more carefully throughout the manuscript, as there are many more instances of this.
2. There are many small mistakes in the main text. PH₃ is not an (oxy)chlorination product of P₄. Phosphate rock is not "V-valent". The phosphoric acid production from the wet process greatly exceeds 9000 tons annually, and making "phosphines" from phosphate in a redox-neutral process makes no sense. Such inaccuracies should be treated more carefully before submission.
3. The discussion needs to be streamlined as there are often redundant parts. In addition, there is a lot of text about the separate yields of products, e.g., the reaction of Cl₂PO₂⁻ with BINOL, but too little discussion about the parts which are scientifically more interesting.
4. The representation of Figure 1 is far too simplified. The oxychlorination of P₄ does not lead to PH₃ – this is simply wrong. Also, the pioneering works by Cummins and Weigand need more credit at this stage. The mechanochemical approach by Cummins is not even mentioned.
5. Regarding P₄ activation: the work by Weigand (Nat. Chem. 2022) and L. L. Liu (JACS, 2022) is missing.
6. Hydride/acetylide reference is incorrect.
7. The abbreviation "Fpyr" is never explained although the addition of this catalyst seems to be essential for the formation of the anion Cl₂PO₂⁻.
8. TBACl is claimed to be a "phase transfer catalyst" but seems to be used stoichiometrically, rendering it effectively just an

in situ ion exchange.

9. Figure 2 lacks any structure. There is poor resolution of the NMR spectra, and the coupling for $F_2PO_2^-$ makes no sense. It is not even discussed in the main text. What is the point of Figure 2g, and why is this important? There is no explanation for it in the main text.

10. The main compound ($PO_2Cl_2^-$) is poorly discussed, with no literature references on prior syntheses, uses, or characterization data such as elemental analysis.

11. Intermediates are insufficiently discussed. The dedicated section in the SI has no useful information and is unintelligibly structured.

12. The recycling of cyanuric acid to recover the chloride is briefly mentioned, but the given reference does not address this question. When the authors claim that it can be recycled, they need to provide suitable data for it. This is essential.

13. Control experiment for the moisture stability of $PO_2Cl_2^-$? As it is called "bench-stable", what exactly does this mean? Can the authors dissolve it in water? Is it only stable to atmospheric moisture?

14. The discussion about the reactivity of the compounds $PO_2Cl_2^-$ and $PO_2F_2^-$, including the bond lengths of P-Cl and P-F bonds, makes no sense at all. It is rather the thermodynamic stabilization of the P-F bonds in this compound that makes it unreactive. This compound is a thermodynamic sink in the degradation of the PF_6^- anion and is very often discussed in publications related to battery research.

15. Why is the P-F reaction even part of the manuscript? It's not substantially adding anything, as all results are preliminary. Stating that the chemistry of this compound ($PO_2F_2^-$) is currently being investigated is just not sufficient.

16. Some design decisions for the reactions are hardly explained. The use of "Fpyr" should at least be mentioned before the mechanistic part. Why is $Na_2S_2O_3$ used in the alcoholysis reactions?

17. What is the reason behind alkynylphosphinates being the only C-centered examples? These derivatives seem to be a niche and not of great interest. What about lithium or magnesium organyls such as BuLi or Grignard reagents for the formation of alkyl or aryl substituted phosphinates?

18. What is the point of the "interactions of phosphate with the TBA cation"? This is not sufficiently explained.

19. Regarding the DFT calculations, it looks like the reaction was calculated to be endergonic overall. The authors use DMF in their calculations and the key intermediate is the Vilsmeier reagent. To bolster the mechanistic assumption, the reaction needs to be carried out using the (commercially available) Vilsmeier reagent. This would be a strong indication for the postulated mechanism. Are there any variable temperature NMR experiments or different stoichiometries of the activating reagents in order to observe any intermediates in NMR spectra?

20. The title of this work needs to be reconsidered. It seems that the authors just combined a few buzzwords to make it sound good or saleable.

21. I was not able to check the supporting information in detail, however, it is full of technical errors, inconsistencies, and typos.

Reviewer #2

(Remarks to the Author)

Direct conversion of phosphates into phosphorylation reagents remains a significant challenge, currently, only a few related methods have been reported. In this manuscript, Quan and coworkers reported a simple and efficient method for direct conversion of commercially cheap phosphate salts into the phosphorylation reagents $[TBA][PO_2X_2]$ ($X = Cl, F$), which can serve as stable and versatile precursors for synthesizing P(V)-containing species, through a redox-neutral process by using cyanuric chloride and 1-formylpyrrolidine in the presence of a tetrabutylammonium chloride at ambient conditions; further, the mechanistic insights, including controlled experiments and computational studies, as well as the application potential of the resulting phosphorylation reagents in the synthesis of organophosphorus compounds were also well demonstrated. The results presented in this manuscript are interesting and practical, providing a new method for converting stable, cheap and readily available inorganic phosphates into high value-added organophosphorus compounds. The research work is in-depth, systematic and detailed, and the research results are some innovative. Furthermore, the characterization of compounds is perfect and the analysis and discussion are reasonable. Therefore, I feel that this contribution may be accepted for publication in *Natural Communications* after minor revision.

1. Please briefly describe the role of additive $Na_2S_2O_3 \cdot 5H_2O$ in the phosphorylation reaction between 1a and phenols and benzyl alcohols

2. In page 13, the sentence of "Furthermore, IntD engages with DMF to cyclize into A, with the concomitant production of CA." should be deleted.

3. In Fig. 2, pictures a – c are not clear enough, and need to be improved.

4. Table S2, entries 11 and 12: the chemical formula of "P" source needs to be corrected.

5. In the title of Figs. S5 and S6, the deuterium reagent needs to be checked.

6. In Table S21, the corresponding spectra of DMF is missing.

7. Language polishing measures are necessary, such as "the versatility and utility of 1a reagents" (Page 7). Moreover, manuscript writing deserves further simplification.

Version 1:

Reviewer comments:

Reviewer #1

(Remarks to the Author)

The authors have made substantial revisions to the manuscript, addressing mainly my previous comments, which has improved the overall quality of the work. However, I want to be clear that I still regard this as an average paper that only partially meets the sustainability claims outlined in the introduction.

Since there seems to be a general consensus to move forward with publication, I have just two final suggestions for correction:

1. In Figure 2f, the coupling pattern in the ^{31}P spectrum does not match the assigned $[\text{PO}_2\text{F}_2]^-$ anion. The spectrum shows a quintet rather than the expected triplet, and the authors should clarify this discrepancy.
2. The newly added section on cyanuric chloride recycling also raises some concerns. The use of POCl_3 in the chlorination step appears contradictory to the study's broader sustainability claims. Frankly, this step undermines the argument for replacing the conventional process with a more sustainable alternative, as POCl_3 itself is derived from the oxychlorination of white phosphorus.

Apart from these points, I can support publication of this work after addressing these technical inconsistencies and performing a final language polish.

Reviewer #2

(Remarks to the Author)

The authors have made significant improvement on the experimental part and writing of this work based on our suggestions. Furthermore, the additional experiments, replacement of references, improved supplementary information and a clearer discussion of the results, which were suggested by another reviewer, largely improved the quality of this work as well. The experimental data provided could fully support their conclusions. In my opinion, the manuscript in its current form may be accepted to publish.

Response to reviewers' comments and revised details

Response to Reviewers' comments:

Reviewer #1 (Remarks to the Author):

This manuscript draft by Quan and colleagues describes the activation of various phosphate sources to dichlorophosphate anions, enabling subsequent functionalization with a range of O-, S-, N-, and C-centered nucleophiles under cleavage of the P-Cl bonds. This is achieved with cyanuric chloride as a chlorinating agent, 1-formylpyrrolidine as a catalyst, and tetrabutylammonium cations for increased solubility of the phosphates, identified as the preferred set of reagents from an extensive reaction screening. The authors further expand this methodology to include fluoride as an alternative leaving group. Additionally, mechanistic studies on the chlorinating activation step are presented.

Overall, I think that this work is of great interest to the scientific community in principle. The synthetic methodology is very simple and opens up an attractive alternative route to many organophosphorus compounds, which is underpinned by a large scope of exemplary target compounds. However, the scholarly representation of these results in the manuscript in its current form is insufficient and not suitable for publication. Because of this, I cannot recommend publication of this manuscript in this journal (or any other journal) in its current form and would advise the authors to significantly revise both their manuscript and supporting information, as many of these issues are quite obvious and could have been resolved long before submission. I have listed some exemplary concerns below.

In addition to this, I would like to leave one critical comment on pre-print publications because this work is already online (<https://www.researchsquare.com/article/rs-4564335/v1>) with a DOI number (<https://doi.org/10.21203/rs.3.rs-4564335/v1>). Publishing in pre-print journals without any peer review is dangerous, and this trend is, in my opinion, just the wrong direction but of course my personal view! This current work is unfortunately a showcase for it,

as the scholarly representation and the conclusions derived from the results are not adequate and not by any chance suitable for publication.

Response: Thank you very much for your thorough and insightful review of our manuscript. We sincerely appreciate your recognition of the potential impact and relevance of our work to the scientific community, especially regarding the simplicity and wide applicability of our synthetic methodology for organophosphorus compounds. We fully agree with your assessment and are committed to addressing these issues comprehensively to enhance the quality and clarity of our manuscript.

According to your suggestion, we thoroughly revised both the manuscript and the supporting information, to provide a more rigorous and detailed explanation, incorporating supplementary experimental data and offering a clearer discussion of the results. Below, we provide detailed responses to each of your insightful concerns, along with the specific changes implemented to address them.

Comment 1. The references given in the manuscript, especially in the introduction, are in many cases not appropriate for the statements they are meant to support. To give an example, the very first reference to support the importance of phosphorus in biological systems is a medicinal review on the role of phosphate in kidney diseases. Using such a specific reference for a very general point seems completely arbitrary to me, and the authors should choose their references more carefully throughout the manuscript, as there are many more instances of this.

Response: According to your suggestion, we replaced the initial reference regarding the importance of phosphorus in biological systems with a more relevant and comprehensive source. The original references 1-5 were cited (*J. Chem. Educ.* (1933); *Science* **235**, 1173-1178 (1987); *Nature* **406**, 31, 33 (2000); *ACS Cent. Sci.* **7**, 1473-1485 (2021); *Nat. Geosci.* **16**, 399–409 (2023).) We also modified other instances in the manuscript (for example, added references 51-52 discussed on phosphodichloridate: *Chem. Res. Toxicol.* **16**, 350-356 (2003); *J. Org. Chem.* **77**, 5829-5831 (2012).).

Comment 2. There are many small mistakes in the main text. PH₃ is not an (oxy)chlorination product of P₄. Phosphate rock is not “V-valent”. The phosphoric acid

production from the wet process greatly exceeds 9000 tons annually, and making “phosphines” from phosphate in a redox-neutral process makes no sense. Such inaccuracies should be treated more carefully before submission.

Response: The errors mentioned in the article have been carefully reviewed and corrected.

We are grateful for that and have corrected this in the revised draft (in page 2, the first paragraph, “..... Developed synthesis of OPCs relies on bulk chemicals such as PCl_3 , PCl_5 , POCl_3 , P_2O_5 , and PH_3 , which are produced by the (oxygen)chlorination reaction of white phosphorus (P_4) or by the hydrogenation of P_4 (thermal process)¹³⁻¹⁸ (Fig. 1a, b). These bulk chemicals present a significant environmental hazard due to their nature as dangerous volatile liquids, gas and involved tedious work-up processes. Recent advancements have been promised, with the development of safer, low-energy methods for synthesizing P_4 ^{19, 20}.”)

The term "V-valent" for phosphate rock has been corrected. Phosphate rock typically contains phosphorus in the +5 oxidation state, which is more accurately referred to as pentavalent rather than V-valent. In the revised draft, we have made this correction. We acknowledge the error regarding the scale of the literature reviewed and have updated this information in the revised draft. “9,000 metric tons” was revised as “90 million tons”.

The sentences in page 2 “Phosphorus in nature mainly exists in the form of V-valent phosphate rock minerals, of which approximately 9,000 metric tons are used for industrial phosphoric acid production every year (wet process)..... under redox-neutral conditions is of great significance in the field of chemical research.” were revised as “Phosphorus predominantly occurs in nature as phosphate rock minerals, existing in the most stable +5-oxidation state and widely recognized as pentavalent phosphorus (P(V)). Annually, approximately 90 million tons of these minerals are utilized for the industrial production of phosphoric acid through the wet process. Consequently, the pursuit of efficient and cost-effective conversion techniques for affordable phosphates into reactive and stable phosphorus synthons under redox-neutral

conditions is of paramount importance.”

Comment 3. The discussion needs to be streamlined as there are often redundant parts. In addition, there is a lot of text about the separate yields of products, e.g., the reaction of Cl_2PO_2^- with BINOL, but too little discussion about the parts which are scientifically more interesting.

Response: According to your suggestion, we revised the discussion section and consolidated overlapping content and emphasized the key scientific insights to enhance overall clarity and readability. We minimized the detailed descriptions of the separate yields of products and redirected our attention toward the broader implications and significance of our findings.

Comment 4. The representation of Figure 1 is far too simplified. The oxychlorination of P_4 does not lead to PH_3 this is simply wrong. Also, the pioneering works by Cummins and Weigand need more credit at this stage. The mechanochemical approach by Cummins is not even mentioned.

Response: According to your suggestion, this statement about PH_3 was corrected. Additionally, discussions on the pioneering works of Cummins and Weigand was included. Figure 1 was revised to accurately reflect the process of phosphates transformation and the pioneering works (Fig. 1c-f). Figure 1 was revised as following:

The following description have been added to the revised manuscript in pages 2-3: In 2018, Cummins..... A significant breakthrough occurred in 2023 when two independent groups, led by Cummins and Weigand, respectively, achieved notable progress in this area. Cummins et al.⁴⁵ developed the mechanochemical phosphorylation of acetylides with condensed phosphates, introducing a novel conversion strategy for phosphates into OPCs (Fig. 1e). Weigand and colleagues⁴⁶ reported an approach using trifluoromethanesulfonic anhydride (Tf₂O) and pyridine to directly convert P(V) sources into the versatile PO₂⁺ phosphorylation agent (pyridine)₂PO₂[OTf]. Tf₂O is essential in breaking P–O bonds and stabilizing the resulting cationic P(V) center, with the aid of N-donor bases acting as effective electron donors. These methods provide redox-neutral access to a range of value-added P(V) chemicals downstream of low-cost phosphoric acid or other phosphate sources (Fig. 1f).

Comment 5. Regarding P₄ activation: the work by Weigand (Nat. Chem. 2022) and L. L. Liu (JACS, 2022) is missing.

Response: According to your suggestion, we have Thank you for the kind

comments and good suggestions. We greatly value these literatures and cited the references (*J. Am. Chem. Soc.* **144**, 1517-1522 (2022), *Nat Chem* **14**, 384-391 (2022)) in revised edition (ref. 25 and 26).

Comment 6. Hydride/acetylide reference is incorrect.

Response: The relevant literature was corrected.

Comment 7. The abbreviation “FPyr” is never explained although the addition of this catalyst seems to be essential for the formation of the anion Cl_2PO_2^- .

Response: We provided explanation of the abbreviation “1-formylpyrrolidine (FPyr)” at its first occurrence in the text in page 4. It has been proved by literature and experiments that Fpyr (*Angew. Chem. Int. Ed.* **55**, 10145-10149 (2016), *Chem. Sci.* **10**, 7399-7406 (2019)), as an amide catalyst, plays a vital role in the formation of active Vilsmeier reagents. It is discussed in the mechanism section.

Comment 8. TBACl is claimed to be a “phase transfer catalyst” but seems to be used stoichiometrically, rendering it effectively just an in situ ion exchange.

Response: According to your suggestion, we conducted the reaction of $[\text{TBA}][\text{H}_2\text{PO}_4]$ and TCT by modifying the standard conditions with different amount of TBAC. We also conducted the reaction of K_3PO_4 with TCT using tetramethylammonium (TMA) salts (TMAC, TMAB) instead of TBAC, under the standard conditions. The results indicate the pivotal role of TBA salt in facilitating the chlorination reaction and a stoichiometric amount of TBAC is essential in completing the reaction. Thus, we hypothesized that increasing the solubility and reactivity of phosphate salt can engage in dual ion-pairing and H-bonding interactions between the $[\text{TBA}]^+$ cation and PO_4^{3-} anion (Fig. 1C). The term "phase transfer catalyst" was revised and a reasonable explanation for the role of TBAC in the reaction was added to the revised manuscript (in pages 4 and 6) and the revised Supporting Information (see Supporting Information S4.1.1, Table S20).

Fig. 1C. A suggested mode of enhancing the solubility and reactivity of phosphate with TBAC.

Figure S6 ^{31}P NMR monitoring the effect of TBAC in the reaction for **1a**

Comment 9. Figure 2 lacks any structure. There is poor resolution of the NMR spectra, and the coupling for F_2PO_2^- makes no sense. It is not even discussed in the main text. What is the point of Figure 2g, and why is this important? There is no explanation for it in the main text.

Response: According to your comment, we carefully reviewed the information in Fig. 2, and reconstructed it.

a: We changed the column diagram (Fig. 2b, 2c, 2f) for the range of chlorinating reagents and the range of phosphates for the synthesis of Cl_2PO_2^- and F_2PO_2^- to a clearer and more comprehensive representation.

b: The quality of the NMR spectrum was improved.

c. Fig. 2 provides a clearer way to show the related properties of Cl_2PO_2^- and

F_2PO_2^- . The relevant content is shown in Fig. 2d-2g. Therefore, we discussed in the revised manuscript as follows: “The halogenation conversion of phosphates produced solid reagents $[\text{TBA}][\text{PO}_2\text{Cl}_2]$ and $[\text{TBA}][\text{PO}_2\text{F}_2]$, stable under ambient conditions, unlike toxic gases or moisture-sensitive liquid phosphate halides (PCl_3 , POCl_3 , POF_3 , PSF_3 and PF_5).” (Page 7)

Fig. 2 was revised as following:

Fig. 2 Preparation of P(V)-X reagents.

Comment 10. The main compound (PO_2Cl_2^-) is poorly discussed, with no literature references on prior syntheses, uses, or characterization data such as elemental analysis.

Response: Thank you for the important comment. According to your comment, we

have cited the original references about the synthesis and hydrolysis studies of PO_2Cl_2^- (*Chem. Res. Toxicol.* **16**, 350-356 (2003).; *J Org Chem* **77**, 5829-5831 (2012).) in the revised manuscript (ref. 51 and 52). And the following description have been added to the revised manuscript in page 6 “Generally, phosphoryl dichloride (KPO_2Cl_2) is obtained by hydrolysis of POCl_3 in aprotic solvent under the action of KHCO_3 or K_2CO_3 . The corresponding fluoride (KPO_2F_2) is obtained by fluoride-chloride exchange of KPO_2Cl_2 .” And “After reaction completion, it indicated clean conversion to one new phosphorus containing product that gives a distinct singlet signal at $\delta = -6.77$ ppm (Fig. **2d**), which is consistent with the previously reported result⁵¹.”

And in page 7 “The halogenation conversion of phosphates produced solid reagents $[\text{TBA}][\text{PO}_2\text{Cl}_2]$ and $[\text{TBA}][\text{PO}_2\text{F}_2]$, stable under ambient conditions, unlike toxic gases or moisture-sensitive liquid phosphate halides (PCl_3 , POCl_3 , POF_3 , PSF_3 and PF_5). $[\text{TBA}][\text{PO}_2\text{Cl}_2]$ hydrolyzes rapidly in water to form phosphate acid. However, it exhibits greater stability exposed to air or solved in organic solvents, alongside very slowly dimerizing. In a sealed environment, it remains stable for up to 30 days without any signs of decomposition. Regarding $[\text{TBA}][\text{PO}_2\text{F}_2]$, it exhibits greater stability than PO_2Cl_2^- (Supplementary Information, **Table S11**). It is not only stable in organic solvents but also remarkable for its stability in water. Analysis of P-Cl and P-F bond lengths reveals weaker P-Cl bonds (~ 2.0 Å), which may facilitate simultaneous reactions, contrasts with the thermodynamically stable P-F bonds (~ 1.5 Å). These findings point to the P-X (X =F, Cl) bonded compounds as stable reagents suitable for the synthesis of OPCs (refer to Supporting Information S 2.4).”

Comment 11. Intermediates are insufficiently discussed. The dedicated section in the SI has no useful information and is unintelligibly structured.

Response: During the reaction for synthesis of **1a**, some important intermediates, such as Vilsmeier reagents, zwitterionic intermediate, monochlorophosphate ions and phosphate dimers were detected. These results were discussed in the revised manuscript:

In page 6 “³¹P NMR indicates that the reaction begins with the formation of an intermediate, displaying unique chemical shifts at $\delta = -28$ ppm, which then undergoes

further transformations to product **1a**. Regrettably, isolating – 28 ppm intermediate is not feasible (refer to Supporting Information S.4.1, Table S21).”

In page 12 “We promise another key step in the chlorination process is the activating TCT by FPyr, which yields highly active Vilsmeier reagents essential for the successful completion of the reaction. In the absence of FPyr, the chlorination reaction resulted in only a 30% isolated yield of **1a** (control **d**). We further conducted a scaled-up reaction using the Vilsmeier reagent and [TBA][H₂PO₄], achieving **1a** with an 85% yield (control **e**). In addition, a zwitterionic intermediate **B** in the chlorination reaction of [TBA][H₂PO₄] was detected by HRMS ([M+K]⁺ *m/z* 217.9979) (Fig. 6). The results suggest that the pivotal factor in this reaction is the catalytic function of FPyr, which is responsible for the formation of highly reactive Vilsmeier reagents.”

Comment 12. The recycling of cyanuric acid to recover the chloride is briefly mentioned, but the given reference does not address this question. When the authors claim that it can be recycled, they need to provide suitable data for it. This is essential.

Response: According to your comment, we conducted a recycling experiment for cyanuric acid at a scale of 20 mmol of phosphate. Upon completion of the reaction, we isolated the crude product by washing the insoluble material with dichloromethane (3 × 20 mL), resulting in a white solid with a yield of 96%. Subsequently, we performed a recycling experiment for cyanuric acid under the documentary conditions. The CA was treated with POCl₃ and *N,N*-diethylethylidene under reflux, yielding recycled cyanuric chloride with 76% yield. This information has been included in the support information section S 4.4, titled 'Cyanuric acid recovery'.

Comment 13. Control experiment for the moisture stability of PO_2Cl_2^- ? As it is called “bench-stable”, what exactly does this mean? Can the authors dissolve it in water? Is it only stable to atmospheric moisture?

Response: By consulting the literature, it was found that KPO_2Cl_2 decompose rapidly in water. At higher pH values, the observed hydrolysis rate increases with the increase of hydroxide ion concentration. To compare the effect of TBA cation to the stability of PO_2Cl_2^- , we conducted stable studies of $[\text{TBA}][\text{PO}_2\text{Cl}_2]$ **1a** and $[\text{TBA}][\text{PO}_2\text{F}_2]$ **1c** over various conditions. All the results have been added to the revised Supplementary Information (**Table S11**).

Table S11 Comparison of the stability of **1a** and **1c** under different conditions

1a stability	1a in D ₂ O	0 h					
	1a in CDCl ₃	0-3 d	3-9 d				
	1a in Air	0-3 d	3-7 d				
	1a in closed environment	0-3 d	3-6 d	6-12 d	12-18 d	18-24 d	24-30 d
1c stability	1c in CDCl ₃	0-6 month					
	1c in D ₂ O	0-3 d	3-6 d	6-12 d	12-18 d	18-24 d	24-30 d

stabilization relatively stable decomposition

From the table, it can be indicated that:

- PO_2Cl_2^- hydrolyzes rapidly in water to form phosphate acid. However, it exhibits greater stability in organic solvents, alongside very slowly dimerizing.
- PO_2Cl_2^- decomposes slowly under air, with complete decomposition occurring after 6 days. In contrast, when stored in a sealed environment, it remains stable for up to 30 days without any signs of decomposition.
- Regarding PO_2F_2^- , it exhibits greater stability than PO_2Cl_2^- . It remains stable in organic solvents such as chloroform for up to six months and demonstrates stability in water.

Modify as this experiment and the results were included in the support information S2.1, add Figs S2-4.

Comment 14. The discussion about the reactivity of the compounds PO_2Cl_2^- and PO_2F_2^- , including the bond lengths of P-Cl and P-F bonds, makes no sense at all. It is rather the thermodynamic stabilization of the P-F bonds in this compound that makes it unreactive. This compound is a thermodynamic sink in the degradation of the PF_6^- anion and is very often discussed in publications related to battery research.

Response: We agree your perspective on the thermodynamic stabilization of the P-F bonds and its implications for the reactivity of these compounds. According to your comment, the discussion regarding the reactivity of the compounds PO_2Cl_2^- and PO_2F_2^- , including the bond lengths of P-Cl and P-F bonds, has been streamlined.

Comment 15. Why is the P-F reaction even part of the manuscript? It's not substantially adding anything, as all results are preliminary. Stating that the chemistry of this compound (PO_2F_2^-) is currently being investigated is just not sufficient.

Response: According to your comment, we reassessed the need to include P-F reactions in the manuscript. Nevertheless, we realize that this is only a preliminary discovery. After careful consideration, we decided to delete the relevant part of the manuscript regarding the PO_2F_2^- reaction, as the preliminary results were not enough to support its importance. To improve the integrity and readability of the article, we focus on the discussion of PO_2Cl_2^- .

16. Some design decisions for the reactions are hardly explained. The use of "Fpyr" should at least be mentioned before the mechanistic part. Why is $\text{Na}_2\text{S}_2\text{O}_3$ used in the alcoholysis reactions?

Response: According to your suggestion, the use of "Fpyr" is now mentioned in the mechanistic part. The follows were added to the mechanism section in page 12:

"We promise another key step in the chlorination process is the activating TCT by

FPyr, which yields highly active Vilsmeier reagents essential for the successful completion of the reaction. In the absence of FPyr, the chlorination reaction resulted in only a 30% isolated yield of **1a** (control **d**). We further conducted a scaled-up reaction using the Vilsmeier reagent and [TBA][H₂PO₄], achieving **1a** with an 85% yield (control **e**). In addition, a zwitterionic intermediate **B** in the chlorination reaction of [TBA][H₂PO₄] was detected by HRMS ([M+K]⁺ *m/z* 217.9979) (Fig. 6). The results suggest that the pivotal factor in this reaction is the catalytic function of FPyr, which is responsible for the formation of highly reactive Vilsmeier reagents.”

b. We conducted comparative reactions between **1a** and binaphthol by modifying the standard conditions, with or without presence of Na₂S₂O₃·5H₂O through tracking the reaction using ³¹P NMR. The results have been added to the revised Supplementary Information (**Figure S12**). The results revealed that the addition of Na₂S₂O₃·5H₂O significantly promotes the reaction and enhances the conversion rate. Our current hypothesis is that the enhancement in reaction rate and conversion is attributable to the interaction between S(VI)-atom and Cl⁻ anion. This interaction may act to activate the P-Cl bond, thereby facilitating the alcoholysis process.

The results were discussed in the revised manuscript in page 8 as follows: “The experiments reveal that the addition of Na₂S₂O₃·5H₂O significantly promotes the reaction and enhances the conversion rate (refer to Supporting Information S5). We propose that the acceleration of the alcoholysis reaction by Na₂S₂O₃·5H₂O is attributed to a unique interaction between the S(VI) center and Cl⁻ anion. This interaction weakens the P-Cl bond, making it more susceptible to nucleophilic attack.”

Figure S12 The role of “Na₂S₂O₃·5H₂O” in the phosphorylation of binaphthol using **1a**.

Comment 17. What is the reason behind alkynylphosphinates being the only C-

centered examples? These derivatives seem to be a niche and not of great interest. What about lithium or magnesium organyls such as BuLi or Grignard reagents for the formation of alkyl or aryl substituted phosphinates?

Response: We appreciate your insightful query regarding the scope of substrate. The focus on alkynylphosphinates as the sole C-centered examples was initially driven by the ease of preparation of alkynyl lithium reagents and the successful demonstration of the reaction's feasibility with these substrates.

According to your suggestion, we conducted the reaction of **1a** with other organic reagents, such as organic lithium or magnesium. The corresponding diaryl hypophosphorous acid (**IV-4~IV-7**) was obtained by using aryl lithium reagents, which successfully extended the methodology to include these substrates. However, when attempting to use Grignard reagents for the reaction, we did not observe the formation of the desired products. Instead, the use of BuLi resulted in a mixture without the formation of the target product. The results were discussed in the revised manuscript in page 9 and Fig 3 as follows: “This revealed that **1a** successfully reacts with organolithium reagents forming P-C bonds for the synthesis of di-alkynyl hypophosphorous acid (**IV-1~IV-3**) and diaryl hypophosphorous acid (**IV-4~IV-7**). It is noteworthy that compound **IV-7** exhibits fluorescence at 365 nm. (Fig. 3)”

Comment 18. What is the point of the “interactions of phosphate with the TBA cation”? This is not sufficiently explained.

Response: To verify interactions of phosphate with the TBA cation, we compared the NMR and IR spectra of K_3PO_4 and $\text{K}_3\text{PO}_4\text{-TBAC}$. The experimental results indicate that the addition of TBAC causes the phosphorus and hydrogen spectra to shift to the

high-field region. Furthermore, the infrared spectrum also exhibits a corresponding red shift. These observations suggest that the addition of TBAC may induce a weak interaction with PO_4^{3-} , thereby facilitating the reaction. The details were discussed in the Mechanism section.

The results were discussed in the revised manuscript Fig. 5 as follows: “We hypothesize that improving the solubility of the phosphate salt is essential for the conversion of phosphates to the P(V)-X reagents.In the absence of TBA salts or instead of TBA with TMA salt under optimized conditions, no product was obtained (control **a**, **b**). These observations suggest that the addition of TBAC may mediate ion-exchanging or ion-pairing and H-bonding, and van der Waals forces between $[\text{TBA}]^+$ and PO_4^{3-} , which together activate the phosphate and drive the reaction toward the formation of **1a**⁶⁵⁻⁶⁷.”

Fig. 5 Mechanism verification.

Comment 19. Regarding the DFT calculations, it looks like the reaction was calculated to be endergonic overall. The authors use DMF in their calculations and the key intermediate is the Vilsmeier reagent. To bolster the mechanistic assumption, the reaction needs to be carried out using the (commercially available) Vilsmeier reagent. This would be a strong indication for the postulated mechanism. Are there any variable temperature NMR experiments or different stoichiometries of the activating reagents in order to observe any intermediates in NMR spectra?

Response: We carefully checked the DFT calculations. After careful examination, considering that the H_2PO_4^- salt is used in the experiment, further calculations were performed through the reaction of Vilsmeier reagents and H_2PO_4^- . The results are as following: “Since Vilsmeier anaminium are the most critical intermediate in the chlorination process, DFT calculations mainly focus on the formation of Vilsmeier anaminium (**Int1**) and the formation of key monochloro substituted **Int4**. In addition, we also considered the key **Int2** and **Int3**. The **Int1** is obtained with an energy barrier of 24.3 kcal/mol (**TS(1,2)**). However, the formation of **Int4** only requires very low energy (only 4.6 kcal/mol). Subsequently, **Int4** was further chlorinated to obtain the target product **1a**.”(Page 12)

We conducted the reaction of Vilsmeier reagents and $[\text{TBA}][\text{H}_2\text{PO}_4]$ at a scale of 10 mmol to validate proposed mechanism It formed chlorinated product **1a**, with a yield of 85%. This part of the experiment was added to the Supporting Information S 4.3. titled 'Synthesis of **1a** using Vilsmeier reagent'.

Furthermore, control experiments were carried out, which has been added to the supporting information S4.1. The zwitterionic intermediate **B** and monochlorination product **C** $[\text{M}+\text{H}]^+$ (m/z 114.9353) were detected by high-resolution mass spectrometry (HRMS). ^1P NMR indicates that the reaction begins with the formation of an intermediate, displaying unique chemical shifts at $\delta = -28$ ppm, which then undergoes further transformations to product **1a**. Regrettably, isolating -28 ppm intermediate is not feasible (refer to Supporting Information S.4.1, Table S21). (Cummins: *J. Am. Chem. Soc.* **145**, 6045-6050 (2023), *J Am Chem Soc* **141**, 6375-6384 (2019)).

Fig. 6 Possible mechanism

Comment 20. The title of this work needs to be reconsidered. It seems that the authors just combined a few buzzwords to make it sound good or saleable.

Response:

We revised title in the updated manuscript as: **Direct conversion of various phosphate sources to a versatile P-X reagent [TBA][PO₂X₂] via redox-neutral halogenation**

21. I was not able to check the supporting information in detail, however, it is full of technical errors, inconsistencies, and typos.

Response:

We apologize for any technical errors, inconsistencies, and typos that may have been present. In response to your comment, we conducted a thorough review of the supporting information to identify and correct any errors.

Reviewer #2 (Remarks to the Author):

Direct conversion of phosphates into phosphorylation reagents remains a significant challenge, currently, only a few related methods have been reported. In this manuscript, Quan and coworkers reported a simple and efficient method for direct conversion of commercially cheap phosphate salts into the phosphorylation reagents [TBA][PO₂X₂] (X = Cl, F), which can serve as stable and versatile precursors for synthesizing P(V)-containing species, through a redox-neutral process by using cyanuric chloride and 1-formylpyrrolidine in the presence of a tetrabutylammonium chloride at ambient conditions; further, the mechanistic insights, including controlled experiments and computational studies, as well as the application potential of the resulting phosphorylation reagents in the synthesis of organophosphorus compounds were also well demonstrated. The results presented in this manuscript are interesting and practical, providing a new method for converting stable, cheap and readily available inorganic phosphates into high value-added organophosphorus compounds. The research work is in-depth, systematic and detailed, and the research results are some innovative. Furthermore, the characterization of compounds is perfect and the analysis and discussion are reasonable. Therefore, I feel that this contribution may be accepted for publication in *Natural Communications* after minor revision.

Response: Thank you for your thorough review and encouraging comments regarding our manuscript. We are delighted that you view our research as innovative and appreciate our proposed method for the direct conversion of phosphate into phosphorylation reagents. Your feedback is invaluable in helping us enhance the quality

of our paper.

In response to your insightful comments and suggestions, we have meticulously revised the manuscript to address your concerns. Below, we provide a detailed account of our specific responses to your feedback. We hope that these revisions will align with your expectations and improve the clarity and scientific rigor of our manuscript.

Once again, we sincerely appreciate your recognition and support of our work, and we eagerly await your further feedback.

Comment 1. Please briefly describe the role of additive $\text{Na}_2\text{S}_2\text{O}_3 \cdot 5\text{H}_2\text{O}$ in the phosphorylation reaction between **1a** and phenols and benzyl alcohols

Response: According to your suggestion, we conducted comparative reactions between **1a** and binaphthol by modifying the standard conditions, with or without presence of $\text{Na}_2\text{S}_2\text{O}_3 \cdot 5\text{H}_2\text{O}$ through tracking the reaction using ^{31}P NMR. The results have been added to the revised Supplementary Information (**Figure S12**). The results revealed that the addition of $\text{Na}_2\text{S}_2\text{O}_3 \cdot 5\text{H}_2\text{O}$ significantly promotes the reaction and enhances the conversion rate. Our current hypothesis is that the enhancement in reaction rate and conversion is attributable to the interaction between S(VI)-atom and Cl^- anion. This interaction may act to activate the P-Cl bond, thereby facilitating the alcoholysis process.

We updated our response in the revised manuscript in page 8.

Comment 2. In page 13, the sentence of “Furthermore, IntD engages with DMF to cyclize into A, with the concomitant production of CA.” should be deleted.

Response: We considered your suggestion and removed the corresponding reaction path in the updated manuscript. We believe this change will improve the clarity of the manuscript. Thank you for your careful review!

Comment 3. In Fig. 2, pictures a – c are not clear enough, and need to be improved.

Response: We reintegrated Figure 2 in the updated manuscript. We enhanced the resolution and overall quality of images to ensure they are clear and informative. Your input is invaluable in improving the quality of our manuscript, and we appreciate your attention to detail.

Figure 2 was revised in the revised manuscript (page 5).

Comment 4. Table S2, entries 11 and 12: the chemical formula of "P" source needs to be corrected.

Response: We appreciate your attention to detail and your assistance in improving the quality of our work. The chemical formula as suggested to ensure accuracy in our manuscript was corrected. The specific revisions are as follows: Table S2, entry11: $\text{Ca}(\text{H}_2\text{PO}_4)_2$; entry12: $\text{Ca}(\text{H}_2\text{PO}_4)_2 \cdot \text{H}_2\text{O}$.

5. In the title of Figs. S5 and S6, the deuterium reagent needs to be checked.

Response: Figs. S5 and S6 erroneously changed the deuterated reagent (CDCl_3) to (DMSO-D_6),

Comment 6. In Table S21, the corresponding spectra of DMF is missing.

Response: Thank you for your detailed review of our manuscript. We made a mistake in original manuscript instead of PFyr with DMF, which has corrected in the revised Supplementary Information (Table S21, Figure S7).

Comment 7. Language polishing measures are necessary, such as “the versatility and utility of 1a reagents” (Page 7). Moreover, manuscript writing deserves further simplification.

Response:

We take your suggestions for language polishing and simplifying your writing very seriously. Here's what we responded to your comments:

We have done a thorough linguistic polish of the manuscript including the phrase "the versatility and utility of **1a** reagents" that you mentioned. We simplified complex sentences, removed redundant information, and made sure that the topic of each paragraph was clearer.

We believe that these changes will significantly improve the quality of the manuscript. Thank you again for your support and guidance in our work.

Response to reviewers' comments and revised details

We would like to express our sincere gratitude to the two reviewers for their invaluable feedback and constructive comments on our manuscript, as well as for the opportunity to revise it. We have carefully considered each suggestion and made the necessary revisions. Below, we provide a detailed response to each comment.

Response to Reviewers' comments:

Reviewer #1 (Remarks to the Author):

The authors have made substantial revisions to the manuscript, addressing mainly my previous comments, which has improved the overall quality of the work. However, I want to be clear that I still regard this as an average paper that only partially meets the sustainability claims outlined in the introduction. Since there seems to be a general consensus to move forward with publication, I have just two final suggestions for correction:

Comment 1. In Figure 2f, the coupling pattern in the ^{31}P spectrum does not match the assigned $[\text{PO}_2\text{F}_2]^-$ anion. The spectrum shows a quintet rather than the expected triplet, and the authors should clarify this discrepancy.

Response:

Thank you for your review of our manuscript and your valuable comments. We acknowledge that the presence of a quintet instead of the anticipated triplet raises an important question. In the ^{31}P spectrum, we have observed both quintet and triplet. Now, we again investigated potential causes for this discrepancy, including solvent effects and sample purity, which may influence the coupling constants. ^{31}P spectrum was reconducted on further purified samples using CDCl_3 and Acetone- d_6 as solvents, respectively, and both solvents resulted in triplet. This suggests that the previously observed phenomenon may be related to the sample purity. We have made the necessary adjustments to Figure 2f and the representation of $[\text{PO}_2\text{F}_2]^-$ in the manuscript:

^{31}P NMR spectra in CDCl_3 of $[\text{TBA}][\text{PO}_2\text{F}_2]$

^{31}P NMR spectra in $\text{Acetone-}d_6$ of $[\text{TBA}][\text{PO}_2\text{F}_2]$

Comment 2. The newly added section on cyanuric chloride recycling also raises some concerns. The use of POCl_3 in the chlorination step appears contradictory to the study's broader sustainability claims. Frankly, this step undermines the argument for replacing the conventional process with a more sustainable alternative, as POCl_3 itself is derived from the oxychlorination of white phosphorus.

Response:

We fully appreciate your concerns regarding the use of POCl_3 in the chlorination step of cyanuric chloride recycling and have given this matter considerable thought. In industrial production, TCT is prepared by reaction of NaCN and Cl_2 . The cyanuric acid

(CA) generated after the TCT reaction is a non-polluting, green reagent, suitable for various fields such as food additives. The method proposed in this paper utilizes chlorinating agents to regenerate and recycle TCT, aiming to develop a TCT-based chlorinating agent recycling method in order to achieve efficient and sustainable conversion and utilization of phosphates.

Considering your concerns and the non-sustainable nature of the source and use of POCl₃, we attempted to replace it with other chlorinating agents. Therefore, we explored alternative chlorination reagents to reduce reliance on POCl₃. We conducted experiments using oxalyl chloride, thionyl chloride, and triphosgene as chlorination agents. The results indicated that thionyl chloride could serve as an alternative chlorine reagent under unoptimized conditions, achieving a CA recovery yield of 57%. We have made corresponding updates in the Supporting Information section S 4.4, under the title "Cyanuric Acid Recovery."

A solution of the [TBA][H₂PO₄] (33.9 g, 20 mmol), and TCT (2.8 g, mmol, 76 mol%) and FPyr (0.096 mL, 1 mmol, 5 mol%) in CH₃CN (20 mL) was stirred at room temperature for 12 hours. The precipitate was obtained by filtration, washed with dichloromethane, and then recrystallized to obtain cyanuric acid with a yield of 96 %. SOCl₂ (1.14 mL, 3.3 eq) was slowly added to a mixture of cyanuric acid (387 mg, 3 mmol) and N,N-diethylenilide (0.75 mL, 1.6 eq) and reacted at 30 °C for 6 h to obtain recovered cyanuric chloride in a yield of 57%. This information has been included in the support information section S 4.4, titled' Cyanuric acid recovery'.

Entry	CA	"Cl" reagent	Additive	T/°C	solvent	Yield
1	3 mmol	SOCl ₂ (3.3 eq)	N,N-Diethylaniline	30	CH ₂ Cl ₂	57
2	3 mmol	BTC (1.5 eq)	N,N-Diethylaniline	30	CH ₂ Cl ₂	N.R
3	3 mmol	(COCl) ₂ (3.3 eq)	N,N-Diethylaniline	30	CH ₂ Cl ₂	N.R

4	3 mmol	SOCl ₂ (3.3 eq)	N,N-Diethylaniline	30	CH ₃ CN	N.R
5	3 mmol	BTC (1.5 eq)	N,N-Diethylaniline	30	CH ₃ CN	N.R
6	3 mmol	(COCl) ₂ (3.3 eq)	N,N-Diethylaniline	30	CH ₃ CN	N.R
7	18 mmol	POCl ₃ (2.5 eq)	N,N-Diethylaniline	105	\	76
8	3 mmol	(COCl) ₂ (3.3 eq)	DMF	30	CH ₂ Cl ₂	\
9	3 mmol	(COCl) ₂ (3.3 eq)	DMF	30	CH ₃ CN	trace

Apart from these points, I can support the publication of this work after addressing these technical inconsistencies and performing a final language polish.

Response:

Thank you for your valuable feedback on our work. We have thoroughly reviewed the experiments and conclusions in the manuscript to ensure the consistency and accuracy of all data and results.

a. We have further refined the abstract portion of this manuscript.

The abstract “Inorganic phosphates possess significant potential as ideal natural building blocks, in the phosphorylation of organic compounds, including previously challenging substrates.” was revised as “Inorganic phosphates hold significant potential as ideal natural building blocks, The approach enables effective halogenation conversion for various P(V) sources, including orthophosphates, pyrophosphoric acid, Na₃P₃O₉ and P₂O₅. Key advantages of this conversion process include the use of inexpensive and readily available chemicals, the avoidance of high-energy redox reactions, and the generation of a reactive yet stable P(V)-X reagent.”

b. The description of the mechanism section has been reorganized and optimised. The sentences on pages 11-12 “We hypothesize that improving the solubility of the phosphate salt is essential for the conversion of phosphates to the P(V)-X reagents. function of FPyr, which is responsible for the formation of highly reactive Vilsmeier reagents.” were revised as “To gain deeper insights into this redox-neutral chlorination process, a series of mechanistic experiments were conducted. Notably, a soluble guanidine phosphate salt produced the dichlorophosphoryl product **1b** in 86% yield, even in the absence of TBAC (control c).

To further probe the effect of TBAC on the reaction system, K₃PO₄ was used as a

model compound, hydrogen bonding, and van der Waals interactions between [TBA]⁺ and PO₄³⁻, collectively activating the phosphate and driving the formation of product **1a**⁶⁵⁻⁶⁷.

A crucial step in the chlorination process is believed to involve the activation of tricyanuric chloride (TCT) by 1-formylpyrrolidine (FPyr), These results underscore the catalytic role of FPyr in generating the reactive Vilsmeier intermediates that are pivotal for the chlorination process. (control c).”

c. Before the final submission, we conducted a comprehensive linguistic polish of the manuscript to enhance its readability and professionalism.

The sentences on page 4 “Generally, phosphoryl dichloride (KPO₂Cl₂) is obtained fluoride (KPO₂F₂) is obtained by fluoride-chloride exchange of KPO₂Cl₂⁵¹. ” were revised as “Phosphoryl dichloride (KPO₂Cl₂) is typically synthesized by reacting POCl₃ with KHCO₃ obtained through a fluoride-chloride exchange reaction of KPO₂Cl₂⁵¹. ”

The sentences on pages 4-5 “³¹P NMR indicated that the reaction begins with the formation of an intermediate, which is consistent with the previously reported result^{51, 52}. ” were revised as “The formation of [PO₂Cl₂]⁻ (signal, δ = -6.77 ppm), as evidenced by ³¹P NMR spectroscopy of the reaction mixture. isolating the intermediate at -28 ppm is not feasible (refer to Supporting Information S.4.1, Tables S19 and S21). This finding is consistent with previously reported results^{51, 52}. ”

The sentences on page 7 “Stability experiments alongside very slowly dimerizing.” were revised as “Stability experiments indicate that [TBA][PO₂Cl₂] hydrolyzes rapidly in water to form phosphoric acid; however, in organic solvents dimerizing very slowly. ”

The sentences on page 8 “To investigate the compatibility of thiol substrates (**3**), of **1a** with various thiol substrates in the presence of NaOH in THF at rt for 5 minutes (conditions B).” were revised as “By adjusting the optimal conditions to absence of Na₂S₂O₃·5H₂O (conditions B), we successfully synthesized a diverse array of dithiol-phosphonate salts (**II**) through the reaction of **1a** with various thiol substrates (**3**) (Fig. 3).”

d. Further adjustments and revisions have been made to the figures and their

captions throughout the text.

e. The experimental section was thoroughly reviewed, and its description has been standardized for clarity.

Reviewer #2 (Remarks to the Author):

The authors have made significant improvement on the experimental part and writing of this work based on our suggestions. Furthermore, the additional experiments, replacement of references, improved supplementary information and a clearer discussion of the results, which were suggested by another reviewer, largely improved the quality of this work as well. The experimental data provided could fully support their conclusions. In my opinion, the manuscript in its current form may be accepted to publish.

Response:

We sincerely thank you for recommending our manuscript for publication in *Nature Communications* and would like to acknowledge your contribution to improving the quality of our manuscript.